EMBO
Molecular Medicine

# ImmunoPET imaging of Trop2 in patients with solid tumours

Wei Huang[1,6], You Zhang[1,6], Min Cao [ID][2,6], Yanfei Wu[3], Feng Jiao[4], Zhaohui Chu[5], Xinyuan Zhou[1], Lianghua Li [ID][1], Dongsheng Xu [ID][1], Xinbing Pan[1], Yihui Guan[3], Gang Huang[1], Jianjun Liu [ID][1✉], Fang Xie [ID][3✉] & Weijun Wei [ID][1✉]

## Abstract

**Accurately predicting and selecting patients who can benefit from targeted or immunotherapy is crucial for precision therapy. Trophoblast cell surface antigen 2 (Trop2) has been extensively investigated as a pan-cancer biomarker expressed in various tumours and plays a crucial role in tumorigenesis through multiple signalling pathways. Our laboratory successfully developed two $^{68}$Ga-labelled nanobody tracers that can rapidly and specifically target Trop2. Of the two tracers, [$^{68}$Ga]Ga-NOTA-T4, demonstrated excellent pharmacokinetics in preclinical mouse models and a beagle dog. Moreover, [$^{68}$Ga]Ga-NOTA-T4 immuno-positron emission tomography (immunoPET) allowed noninvasive visualisation of Trop2 heterogeneous and differential expression in preclinical solid tumour models and ten patients with solid tumours. [$^{68}$Ga]Ga-NOTA-T4 immunoPET could facilitate clinical decision-making through patient stratification and response monitoring during Trop2-targeted therapies.**

**Keywords** Trop2; ImmunoPET; Nanobody; Companion Diagnostics; Theranostics
**Subject Categories** Cancer; Pharmacology & Drug Discovery

## Introduction

The incidence and mortality rates of patients diagnosed with malignant tumours are rising each year (Siegel et al, 2023). Correspondingly, the number of treatment options available to them is also increasing. However, satisfactory outcomes are yet to be achieved with many of the available treatments. Factors such as tumour drug resistance, heterogeneity, and other factors may coordinately contribute to this outcome (Dagogo-Jack and Shaw, 2017; Gottesman, 2002). Currently, histopathology remains the primary standard for tumour diagnosis. In addition to standard first-line chemotherapeutic agents, utilising targeted therapies and immunotherapies necessitates the detection of biomarkers expressed via immunohistochemistry (IHC) and other detection methods (Saeed et al, 2023). Nonetheless, high biomarker expression does not guarantee treatment success in all patients, while negative biomarker expression has proven effective in some cases (Balar et al, 2017; Daud et al, 2016). All tests that require a biopsy are subject to errors and limitations caused by various factors, such as the deep anatomical location and the spatiotemporal heterogeneity of the tumour, leading to difficulties in invasive tissue collection and inaccurate evaluation of the heterogeneous tumours (Chen et al, 2016; Hegde et al, 2016; Van Allen et al, 2015).

Positron emission tomography (PET) imaging is increasingly essential for tumour diagnosis and efficacy assessment. Immuno-positron emission tomography (immunoPET) is a revolutionary molecular imaging technique that combines the target specificity of antibodies with the inherent sensitivity of PET technology (Ametamey et al, 2008; Rowe and Pomper, 2021). ImmunoPET imaging can provide valuable information on biomarker status throughout the body and treatment response during the treatments (Wei et al, 2020). For instance, Bensch et al employed $^{89}$Zr-labelled atezolizumab (anti-PD-L1 antibody) to predict the potential response to PD-L1 blockade. The study demonstrated that pre-treatment PET signalling, but not the predictive biomarkers from IHC or ribonucleic acid sequencing, predicted the patients' clinical response following PD-L1-targeted immunotherapy (Bensch et al, 2018). Results from a clinical study in human epidermal growth factor receptor 2 (HER2)-targeted therapy indicated that HER2-directed $^{89}$Zr-trastuzumab immunoPET imaging could predict treatment response noninvasively within weeks of treatment initiation. Additionally, early increased uptake of $^{89}$Zr-trastuzumab may identify drug-resistant disease sites before conventional computed tomography (CT) or magnetic resonance imaging (MRI) imaging detects disease progression (Sanchez-Vega et al, 2019). Therefore, the information provided by immunoPET is valuable for optimising the management of human malignancies by noninvasively visualising target dynamics before and after the prescription of molecularly targeted or immunotherapy agents.

Trop2, also known as trophoblast cell surface antigen 2, is a tumour-associated calcium signal transducer (*TACSTD*) gene

[1]Department of Nuclear Medicine, Institute of Clinical Nuclear Medicine, Renji Hospital, School of Medicine, Shanghai Jiao Tong University, 1630 Dongfang Rd, Shanghai 200127, China. [2]Department of Thoracic Surgery, Renji Hospital, School of Medicine, Shanghai Jiao Tong University, Shanghai 200217, China. [3]Department of Nuclear Medicine & PET Center, Huashan Hospital, Fudan University, Shanghai 200040, China. [4]Department of Oncology, State Key Laboratory of Systems Medicine for Cancer, Shanghai Cancer Institute, Renji Hospital, School of Medicine, Shanghai Jiao Tong University, Shanghai 200127, China. [5]Department of Oncology, Huashan Hospital, Fudan University, Shanghai 200040, China. [6]These authors contributed equally: Wei Huang, You Zhang, Min Cao. ✉E-mail: ljjsh@sjtu.edu.cn; fangxie@fudan.edu.cn; wwei@shsmu.edu.cn

family member. The Trop2 gene encodes a transmembrane protein with a single structural domain, ~36 kDa, consisting of 323 amino acids. It contains four potential N-terminal glycosylation sites (Stein et al, 2006). In recent years, accumulating research has shown that Trop2 functions as an unfavourable prognostic biomarker for multiple tumours (Dum et al, 2022). It exhibits high expression in various types of solid tumours such as breast (Zhao et al, 2018), thyroid (Guan et al, 2017), and oral squamous cell carcinomas (Tang et al, 2019) in comparison to other tumour biomarkers. Increased level of Trop2 expression is significantly linked to reduced survival time and unfavourable prognosis (Liu et al, 2022). Studies have shown that abnormal expression of Trop2 in different cultured cancer cells stimulates cell proliferation. Moreover, animal experiments indicate that tumour cells exhibiting high Trop2 expression are generally more tumorigenic. Conversely, treating cells with anti-Trop2 antibodies can prevent tumour growth (Herlyn et al, 1984). In addition to regulating tumour growth, the high expression of Trop2 is closely associated with increased tumour metastasis, as seen in pancreatic (Cubas et al, 2010) and lung cancers (Li et al, 2016). Trop2 facilitates tumour development by mediating multiple signalling pathways and promoting tumour cell growth, proliferation, and metastasis (Lipinski et al, 1981).

Based on the pan-cancer expression of Trop2, it is currently utilised as a target for antibody–drug conjugates (ADCs) in third-line treatment of triple-negative breast cancer, with several ongoing clinical trials exploring the safety profiles and efficacy of novel Trop2-targeted therapies (Bardia et al, 2021a). Nevertheless, effective identification of patients likely to benefit from these treatments remains elusive. It could be advantageous to enhance patients' outcomes and decrease treatment expenses by using the immunoPET imaging technique to noninvasively identify patients with heterogeneous Trop2 expression before treatment with Trop2-targeted ADCs, thereby allowing patient stratification and efficacy prediction. Based on our previous studies (An et al, 2022; Huang et al, 2024; Zhang et al, 2023), this work aimed to develop and characterise novel nanobody-derived immunoPET imaging tracers specific for human Trop2. Following thorough preclinical studies in tumour-bearing mouse models and a healthy beagle dog, a pilot clinical imaging study was carried out to explore the safety profiles and diagnostic value of one of the tracers ([68Ga]Ga-NOTA-T4) in patients with solid tumours. Moreover, head-to-head diagnostic differences of [68Ga]Ga-NOTA-T4 and [18F]-FDG (a tracer reflecting glucose metabolism) were compared in these patients.

# Results

## Characteristics of nanobodies (T4 and T5) and its derivative (ABDT4)

We developed two monovalent nanobodies, i.e., T4 (Fig. 1A−C) and T5 (Fig. 1D−F). The molecular weight of T4 or T5 was almost 15 kDa, as tested by sodium dodecyl sulfate-polyacrylamide gel electrophoresis (SDS-PAGE; Fig. 1A,D). They both possessed more than 95% purity, verified by high-performance liquid chromatography (HPLC; Fig. 1B,E). In addition, the $K_D$ value of T4 and T5 with human Trop2 was 615.8 picomole (pM) and 708.2 pM, respectively (Fig. 1C,F). Binding affinities of T4 and T5 with recombination human Trop2 were robust and stable, a prerequisite for developing molecular imaging or theranostic tracers. Besides, the binding affinities of nanobodies conjugated with the

chelator NOTA (NOTA-T4 and NOTA-T5) with human Trop2 were similar to that of T4 and T5, with the corresponding $K_D$ value of 640.7 pM and 685.4 pM, respectively (Appendix Fig. S1). Therefore, chelator conjugation did not influence the binding affinities of the nanobodies to Trop2. Considering the short biological half-life of monovalent nanobodies, we developed a T4 derivative (ABDT4) by fusing T4 with a potent albumin binder ABD035 (Jonsson et al, 2008). The $K_D$ value of ABDT4 with human Trop2, human serum albumin, and murine serum albumin was 355.50 pM, 33.43 pM, and 0.35 nM, respectively (Appendix Fig. S2). ABDT4 binds tightly to human/murine serum albumin and the resultant prolonged in vivo circulation allows adequate binding to Trop2 on the surface of tumour cells. ABDT4 is used as a blocking agent to prove the specificity of monovalent nanobody tracer in the study.

## [68Ga]Ga-NOTA-T4 and [68Ga]Ga-NOTA-T5 immunoPET imaging in tumour-free mice and cell-derived xenograft (CDX) mice models

We developed two 68Ga-labelled nanobody tracers (i.e., [68Ga]Ga-NOTA-T4 and [68Ga]Ga-NOTA-T5) and evaluated the radio-chemical purity (RCP) before and post purification by PD-10 columns (Appendix Fig. S3). The pre-purification RCP of [68Ga]Ga-NOTA-T4 and [68Ga]Ga-NOTA-T5 was 92% and 86%, respectively. The post-purification RCP reached more than 99% for each of the tracers. We first confirmed the radio-distribution of the two tracers in tumour-free nude mice ($n = 4$/group) at 45 min post-injection (p.i.) of the tracers. Both the tracers were rapidly cleared by the urinary system, with most of the radioactive dose distributed in the bilateral kidneys and to a lesser extent in other tissues/organs (Appendix Fig. S4) due to the molecular weight being less than the clearance cut-off value (60 kDa) of glomerular filtration. The T3M-4 CDX models with high-expression Trop2 were then constructed to assess the diagnostic value of [68Ga]Ga-NOTA-T4 and [68Ga]Ga-NOTA-T5. The high expression of Trop2 in T3M-4 cells was verified in our previous work (Huang et al, 2024). The tumours could be sharply delineated by [68Ga]Ga-NOTA-T4 (Fig. 2A−C) and [68Ga]Ga-NOTA-T5 (Fig. 2D−F) at 45 min p.i., with the corresponding tumour uptake value of 4.60 ± 1.42%ID/g (Fig. 2B) and 2.10 ± 0.59%ID/g (Fig. 2E) on region of interest (ROI) analysis ($n = 4$ per group, $P = 0.018$). Furthermore, [68Ga]Ga-NOTA-T4 and [68Ga]Ga-NOTA-T5 had little difference in radioactivity accumulation in some major organs (like liver, kidneys, lung, or muscle), whereas [68Ga]Ga-NOTA-T4 possessed a higher tumour-to-organ ratio (Appendix Fig. S5). Subsequent biodistribution (Bio-D) results were consistent with the ROI analysis, with the highest radioactivity accumulation organ found in the kidneys, followed by the tumour site (Fig. 2C,F). IHC staining confirmed high Trop2 expression in the T3M-4 xenografts (Fig. 2G,H). Overall, [68Ga]Ga-NOTA-T4 showed a higher target-to-background ratio and better diagnostic ability in preclinical pancreatic cancer models.

## Effect of varying blocking doses on radioactivity uptake of [68Ga]Ga-NOTA-T4

To investigate the influence of different blocking doses on radioactivity uptake in tumours and non-target organs (especially the kidneys), we co-injected different amounts (50–400 μg) of unlabelled T4 as blocking doses and acquired images at two time points (45 min and 2.5 h). The

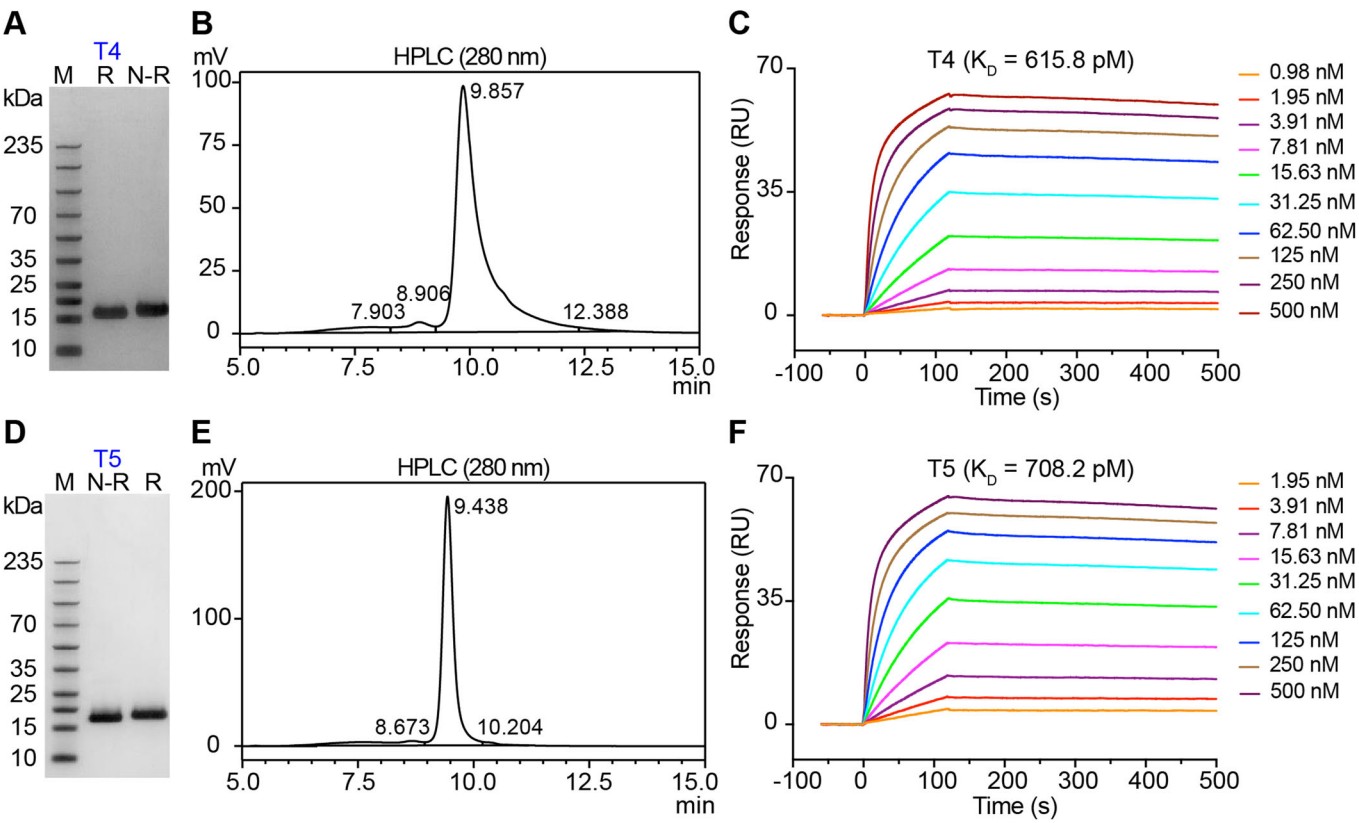

**Figure 1. Characteristics of two nanobodies (T4 and T5) targeting human Trop2.**

(A) SDS-PAGE tested the purity of T4. R: reducing condition; N-R: non-reducing condition. (B) Characterisation of purity of T4 by HPLC. (C) SPR study demonstrated T4 affinity and kinetic towards recombinant human Trop2 protein. The purity of another nanobody, T5, was also determined by SDS-PAGE (D) and HPLC (E). R reducing condition, N-R non-reducing condition. (F) The binding affinity of T5 with human Trop2 protein. Source data are available online for this figure.

maximum intensity projection (MIP) images at 45 min and 2.5 h after co-injection are shown in Fig. 3A–D. The higher the co-injected dose of unlabelled T4, the lower the radioactivity uptake in the tumour and kidneys. Based on the ROI quantitative analysis results, tumour uptake at 45 min p.i. in each group was $2.10 \pm 0.73\%$ID/g (non-blocking group, $n = 5$), $1.63 \pm 0.29\%$ID/g (50 µg-blocking group, $n = 4$), $1.22 \pm 0.38\%$ID/g (200 µg-blocking group, $n = 4$), $0.82 \pm 0.33\%$ID/g (400 µg-blocking group, $n = 4$), respectively (Fig. 3E). The corresponding tumour uptake at 2.5 h p.i. was $2.08 \pm 0.70\%$ID/g (non-blocking group, $n = 5$), $1.43 \pm 0.38\%$ID/g (50 µg-blocking group, $n = 4$), $1.12 \pm 0.38\%$ID/g (200 µg-blocking group, $n = 4$), $0.69 \pm 0.34\%$ID/g (400 µg-blocking group, $n = 4$), respectively (Fig. 3F). The radioactivity uptake at the tumour site did not appear to decrease significantly at 2.5 h in either the non-blocking or blocking groups (Fig. EV1), showing that the tracer was stably and tightly bound to Trop2. As for the radioactivity accumulation in the kidneys, the reduction in renal radioactivity uptake at the higher blocking doses may be attributed to the competition between the tracer and unlabelled "cold" T4 for reabsorption by the renal tubules. With the increase in the co-injection dose of "cold" T4, there is a subsequent reduction in the proportion of the tracer available for reabsorption. The kidney accumulation on the ROI quantitative results at 45 min p.i. was $53.36 \pm 4.92\%$ID/g (non-blocking group, $n = 5$), $45.18 \pm 4.12\%$ID/g (50 µg-blocking group, $n = 4$), $26.35 \pm 9.96\%$ID/g (200 µg-blocking group, $n = 4$), $16.75 \pm 4.43\%$ID/g (400 µg-blocking group, $n = 4$), respectively

(Fig. 3E). The same downward trend in kidney accumulation was observed at 2.5 h p.i. (Fig. 3F). Bio-D results obtained at the end of imaging were similar to those of the quantitative ROI analysis (Fig. 3G). We wondered if there was a specific blocking dose that would optimally balance high tumour uptake with low renal accumulation. A definitive answer was not drawn with the current data by comparing the radioactivity uptake ratios between the tumour and organs in the described blocking groups. As shown in Appendix Fig. S6, the non-blocking group generally had higher tumour-to-organ ratios, including tumour-to-heart, tumour-to-liver, and tumour-to-muscle ratios. And the tumour-to-kidney ratio was not significantly different between the four groups. As for the imaging in the non-blocking group at 45 min and 2.5 h, the tumour-to-organ ratios (TMR, TLR, THR, and TKR) at 2.5 h were higher than that at 45 min, suggesting that delayed imaging at 2.5 h provided better imaging contrast. Hematoxylin–eosin (H&E) staining results demonstrated the cancerous state of the cellular components in each blocking group. IHC of the corresponding tumour tissue sections showed high Trop2 expression levels (Appendix Fig. S7).

## Pre-administration of ABDT4 blocked tumour uptake of [⁶⁸Ga]Ga-NOTA-T4 in tumour-bearing mice models

To further confirm the specific binding of [⁶⁸Ga]Ga-NOTA-T4 to the target Trop2, we administered ABDT4 with an extended

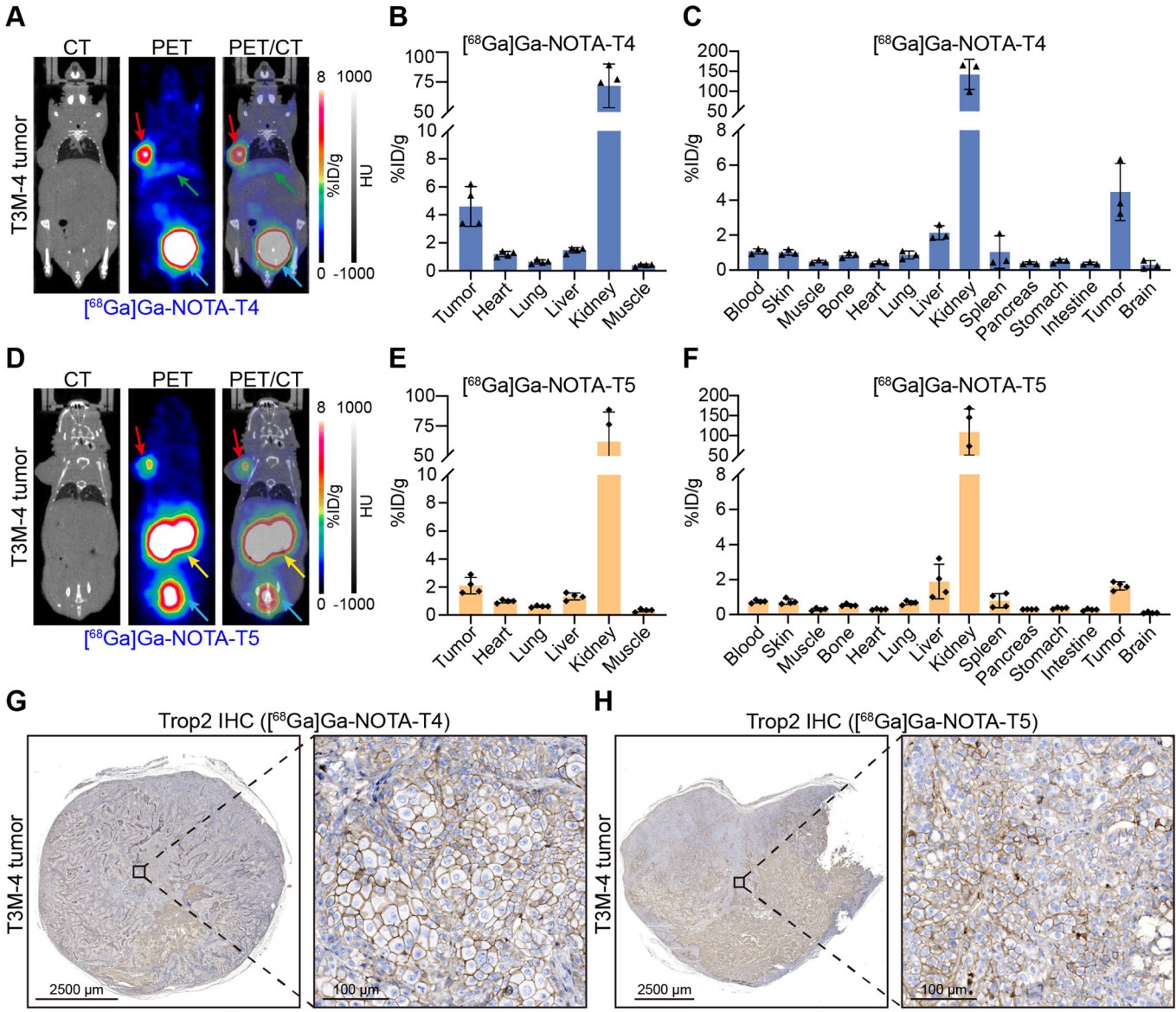

**Figure 2. [⁶⁸Ga]Ga-NOTA-T4 and [⁶⁸Ga]Ga-NOTA-T5 immunoPET imaging in T3M-4 models.**

(A) Representative immunoPET/CT images of [⁶⁸Ga]Ga-NOTA-T4 (n = 4) in T3M-4 tumour xenografts models at 45 min p.i. Red arrows: tumour; green arrows: liver; blue arrows: bladder. (B) ROI analysis of [⁶⁸Ga]Ga-NOTA-T4 (mean ratio ± SD, n = 4). (C) Ex vivo Bio-D data presented the detailed distribution patterns of [⁶⁸Ga]Ga-NOTA-T4 (mean ratio ± SD, n = 3) in the tumours and other major organs/tissues. (D) Coronal immunoPET/CT images of [⁶⁸Ga]Ga-NOTA-T5 in T3M-4 models at 45 min p.i. Red arrows: tumour; yellow arrows: kidneys; blue arrows: bladder. (E) ROI analysis and (F) Bio-D results of [⁶⁸Ga]Ga-NOTA-T5 after imaging (mean ratio ± SD, n = 4). (G, H) Trop2 IHC staining of T3M-4 tumours after the termination of immunoPET imaging. Source data are available online for this figure.

circulation period 48 h beforehand to satisfy the time window for its complete binding to Trop2. The PET signal intensity at the tumour site was considerably low, resulting in unclear visualisation of the tumour's contour (Fig. 4A). Both the ROI and Bio-D data demonstrated that tumour uptake was comparable to that of other non-target organs (Fig. 4B,C). When comparing the tumour uptake in the non-blocking and ABDT4-blocking groups, the tumour uptake value in the ABDT4-blocking group was significantly lower (ROI: 0.56 ± 0.09%ID/g vs. 2.10 ± 0.73%ID/g, P = 0.004, n = 4 and 5; Fig. 4D). Besides, the non-blocking group had better tumour-to-muscle ratio (TMR, 18.23 ± 10.77 vs. 5.09 ± 2.09, P = 0.049), tumour-to-heart ratio (THR, 4.23 ± 1.27 vs. 1.16 ± 0.08,

P = 0.002), tumour-to-liver ratio (TLR, 2.16 ± 0.77 vs. 0.58 ± 0.07, P = 0.005), and tumour-to-kidney ratio (TKR, 0.04 ± 0.01 vs. 0.01 ± 0.00, P = 0.012) than the ABDT4-blocking group (Fig. 4E). The data indicated that [⁶⁸Ga]Ga-NOTA-T4 immunoPET can specifically visualise Trop2 expression level.

## Dynamic distribution patterns of [⁶⁸Ga]Ga-NOTA-T4 in a beagle dog

Before clinical translation, we evaluated the dynamic distribution patterns of [⁶⁸Ga]Ga-NOTA-T4 in a male beagle dog within one hour using state-of-the-art total-body PET/CT scanning

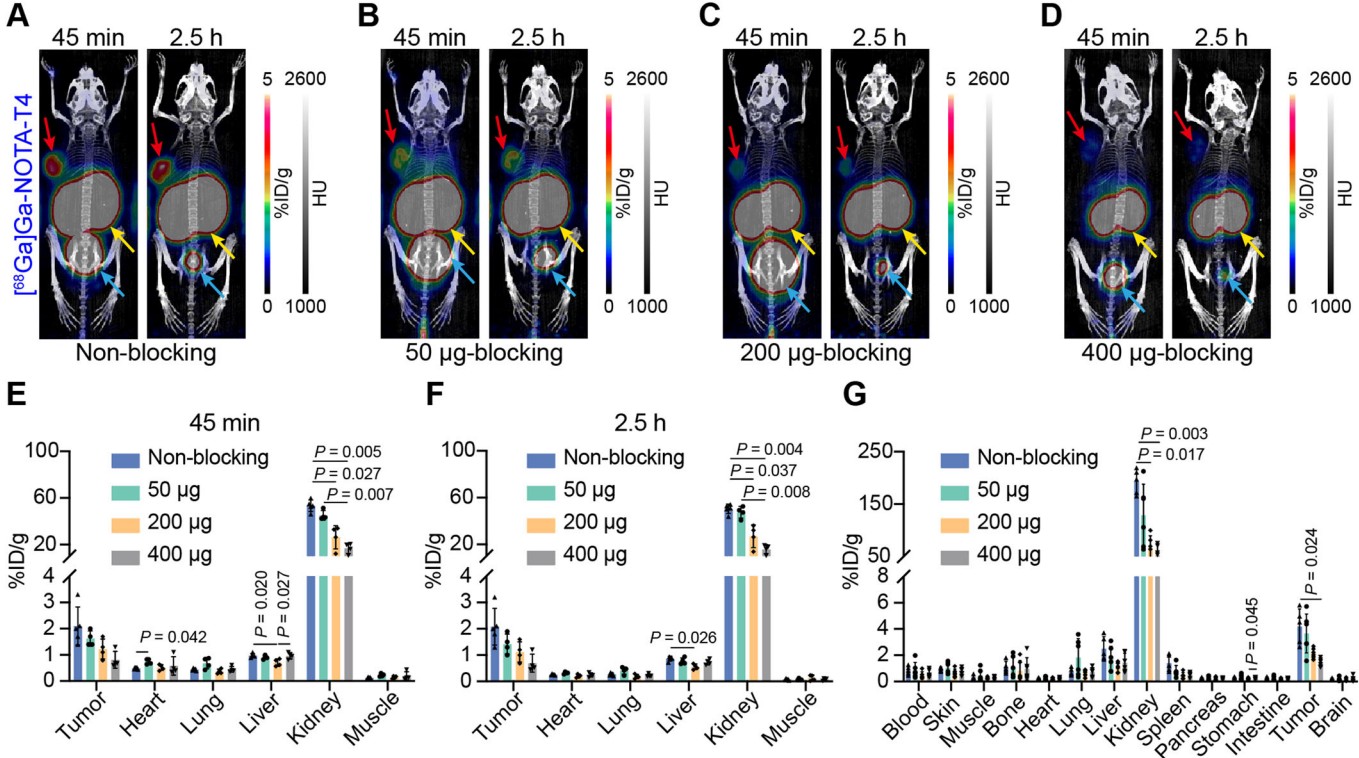

**Figure 3.  Effect of varying blocking doses on radioactivity uptake of [⁶⁸Ga]Ga-NOTA-T4 in tumours and non-target tissues/organs.**

(A–D) Representative MIP images of the non-blocking and three blocking (50, 200, and 400 μg) groups at two time points. Red arrows: tumour; yellow arrows: kidneys; blue arrows: bladder. (E, F) ROI analysis results of the non-blocking and various blocking groups at 45 min (E) and 2.5 h (F). Groups divided into non-blocking group ($n = 5$), 50 μg-blocking group ($n = 4$), 200 μg-blocking group ($n = 4$), and 400 μg-blocking group ($n = 4$). (G) Bio-D data of the non-blocking and various blocking groups after the termination of [⁶⁸Ga]Ga-NOTA-T4 immunoPET imaging. Groups divided into non-blocking ($n = 5$), 50 μg-blocking ($n = 6$), 200 μg-blocking ($n = 6$), and 400 μg-blocking groups ($n = 6$). Two-way ANOVA, mean ratio ± SD. Source data are available online for this figure.

(Movie EV1). Representative whole-body MIP images of [⁶⁸Ga]Ga-NOTA-T4 at different time points (15 s, 25 s, 45 s, 2 min, 5 min, and 60 min) were presented in Fig. 5A. After the bolus injection, [⁶⁸Ga]Ga-NOTA-T4 rapidly pooled into the heart to enter the systemic circulation, followed by the rapid accumulation of the tracer in the kidneys. Time-activity-curve (TAC) analysis showed the dynamic uptake and distribution patterns in the major tissues and organs of the tracer. As shown in Fig. 5B, the highest accumulation of radioactivity was found in the kidneys, followed by the heart (blood pool), then the liver, and lower uptake of radioactivity in all other organs (brain, lung, pancreas, spleen, large intestine, bladder, bone, and muscle) was observed in the first 15 min after the injection of the tracer. After 15 min, the tracer was cleared by the urinary system, and radioactivity uptake of the bladder gradually increased, reaching a peak at 46 min. As shown in Fig. 5C, radioactivity uptake in the kidneys appeared at 16 s after injection, and uptake in the liver began at 23 s. The volume of interest (VOI) curve showed the quantitative $SUV_{mean}$ value in the kidneys (Fig. 5D). The decrease in $SUV_{mean}$ values from 25 to 30 s may be due to a partial volume effect, where the VOI is slightly smaller than the actual size of the organ. In addition, the overall trend of radioactivity uptake in the kidneys showed a gradual increase and levelling off, as evidenced by the results of the overall TAC curve within one hour. The PET/CT scan revealed minimal radioactivity uptake in the major organs and tissues (including bones), except for elevated kidney and bladder accumulation. The background signal was clean. The tracer's good stability and circulation profiles in the beagle dog warranted further translational study in human subjects.

## Head-to-head [¹⁸F]-FDG and [⁶⁸Ga]Ga-NOTA-T4 PET/CT imaging in patients with solid tumours

Currently, there are no clinical-stage molecular imaging tracers specific to Trop2. We utilised one of the nanobody tracers, [⁶⁸Ga]Ga-NOTA-T4, for pilot clinical application for the first time. No adverse reactions were observed in patients during the injection of [⁶⁸Ga]Ga-NOTA-T4 or the 4-h follow-up period. This study recruited ten patients with malignant tumours. Analysis of ROI data in these patients showed that organs/tissues with high tracer uptake in descending order were kidneys, pancreas, and glands (thyroid and salivary glands). In contrast, the other organs/tissues had very low uptake (Appendix Fig. S8). Representative images of three patients (one female and two males) with different types of carcinomas were presented in the current study. One patient had nasopharyngeal carcinoma with multiple systemic metastases, another patient with ovarian cancer surgery history presented with enlarged right hilar and mediastinal lymph nodes for medical imaging assessment, and the

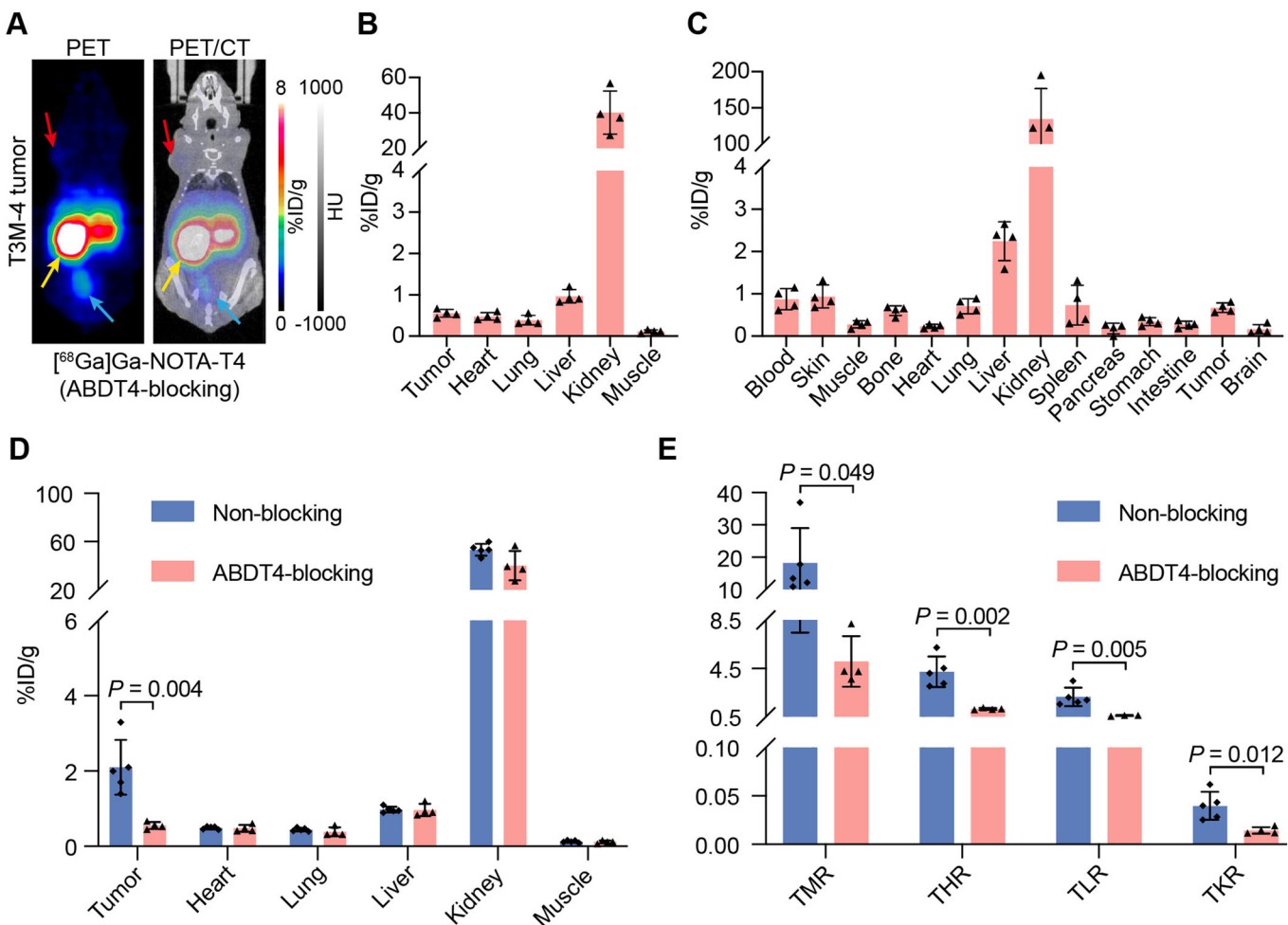

**Figure 4. ABDT4 blocked tumour uptake of [⁶⁸Ga]Ga-NOTA-T4 in tumour-bearing mice models.**

(A) Representative images showed the results of blocking Trop2 on the surface of tumour cells with ABDT4. Red arrows: tumour; yellow arrows: kidneys; blue arrows: bladder. (B) ROI quantitative analysis of [⁶⁸Ga]Ga-NOTA-T4 immunoPET imaging results in the ABDT4-blocking group (mean ratio ± SD, $n = 4$). (C) Ex vivo Bio-D results after termination of [⁶⁸Ga]Ga-NOTA-T4 immunoPET imaging (mean ratio ± SD, $n = 4$). (D) Comparison of radioactivity uptake in tumour and major tissues/organs between the non-blocking ($n = 5$) and ABDT4-blocking groups ($n = 4$). $t$ test, mean ratio ± SD. (E) Comparison of tumour-to-organ ratios between the non-blocking ($n = 5$) and ABDT4-blocking groups ($n = 4$). $t$ test, mean ratio ± SD. TMR tumour-to-muscle, THR tumour-to-heart, TLR tumour-to-liver, TKR tumour-to-kidney. Source data are available online for this figure.

third patient had small-cell lung cancer with multiple systemic metastases. Before undergoing [⁶⁸Ga]Ga-NOTA-T4 immunoPET/CT imaging, all the patients completed traditional [¹⁸F]-FDG PET/CT examinations. Overall, these three patients had lower background activity with [⁶⁸Ga]Ga-NOTA-T4 compared to [¹⁸F]-FDG, since [¹⁸F]-FDG has noticeable physiological uptake in several organs/tissues such as brain, muscle, intestine, and stomach as well as in inflammatory lymph nodes across the body.

Figure 6 displays [¹⁸F]-FDG (A–C) and [⁶⁸Ga]Ga-NOTA-T4 (D–F) PET/CT examination findings in a patient with nasopharyngeal carcinoma suffering from right parapharyngeal space lymph node, right neck lymph node, bone, and liver metastases at diagnosis. On the initial [¹⁸F]-FDG PET/CT examination for staging, the primary tumour (Fig. EV2A) at the right lateral wall of the nasopharynx accompanied by multiple metastases at the right parapharyngeal space lymph node (Fig. EV2B), right neck lymph node (Fig. EV2C), multiple bone metastases (C3 vertebrae, T3 vertebrae (Fig. 6A), left third lateral

rib, right pubis, and left ilium) and liver metastases (Fig. 6C) as well as a suspicious right axillary lymph node metastasis (Fig. EV2D) was reported. The primary tumour was controlled after systemic therapy, including radiotherapy, chemotherapy, and immunotherapy (anti −PD-1 antibody tislelizumab), while other metastatic lesions remained. The patient was then subjected to [⁶⁸Ga]Ga-NOTA-T4 immunoPET imaging to determine Trop2 expression in the metastatic lesions. Apart from the known bone metastasis at T3 vertebrae (indicated by orange arrows) on [¹⁸F]-FDG PET (Fig. 6A) and [⁶⁸Ga] Ga-NOTA-T4 immunoPET (Fig. 6D) imaging, a new bone metastasis at T10 vertebrae (indicated by yellow arrows) occurred on [⁶⁸Ga]Ga-NOTA-T4 immunoPET imaging (Fig. 6E), which was not evident in previous [¹⁸F]-FDG PET imaging (Fig. 6B). Multiple liver metastases (indicated by blue arrows) on [¹⁸F]-FDG (Fig. 6C) PET/CT examination were also evident on [⁶⁸Ga]Ga-NOTA-T4 (Fig. 6F) immunoPET/CT imaging with a median $SUV_{max}$ of 7.6, ranging from 5.6 to 8.9. Histopathological examination of the biopsied liver nodule

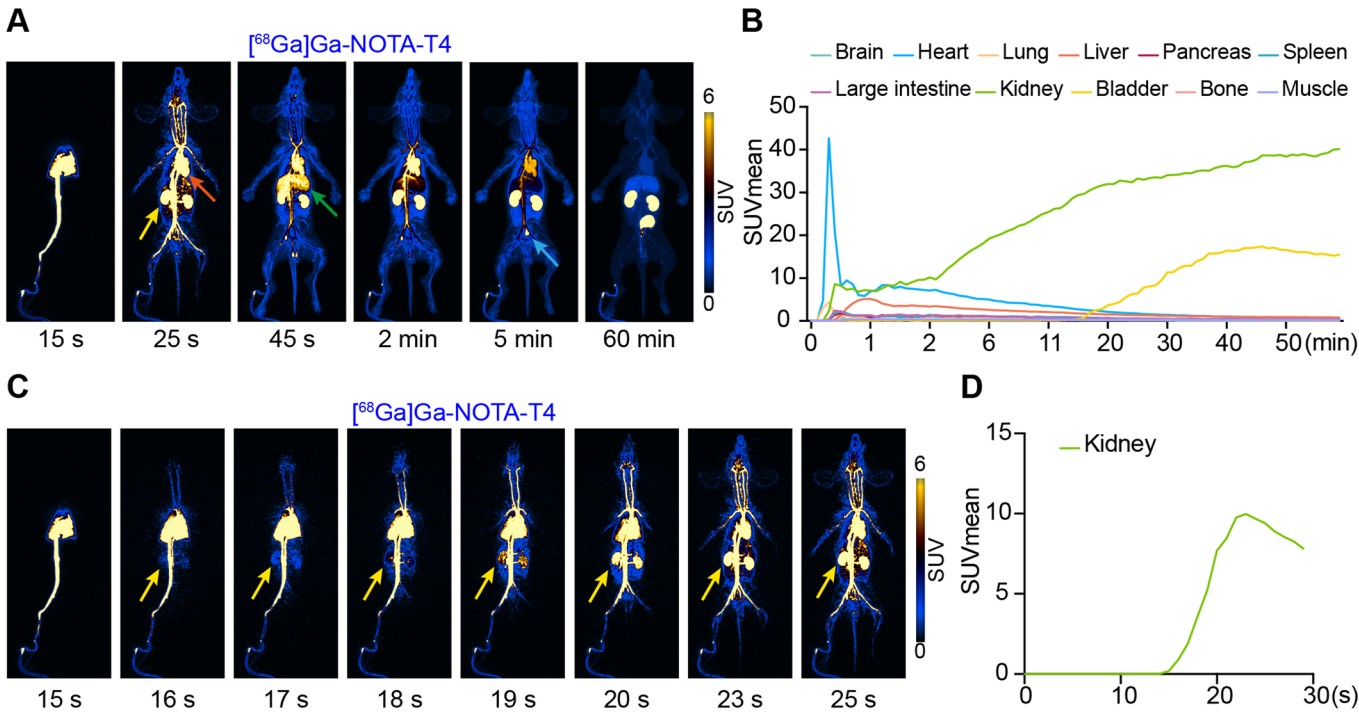

**Figure 5. Dynamic [⁶⁸Ga]Ga-NOTA-T4 immunoPET imaging in an adult male beagle dog.**

(A) Whole-body maximum intensity projection (MIP) images of [⁶⁸Ga]Ga-NOTA-T4 immunoPET imaging at different time points (15 s, 25 s, 45 s, 2 min, 5 min, and 60 min) in a beagle dog. Orange arrow: heart; green arrow: liver; yellow arrow: kidneys; blue arrows: bladder. Scale bar: 0–6. (B) TAC curves of [⁶⁸Ga]Ga-NOTA-T4 in the major tissues/organs, including the brain, heart, lung, liver, pancreas, spleen, large intestine, kidneys, bladder, bone, and muscle. (C) MIP images of [⁶⁸Ga]Ga-NOTA-T4 imaging at 15–20 s, 23 s, and 25 s time points present the process of kidney accumulation. Yellow arrows: kidneys. Scale bar: 0–6. (D) VOI curve of [⁶⁸Ga]Ga-NOTA-T4 in the kidney during 30 s. Source data are available online for this figure.

confirmed metastasis from poorly differentiated nasopharyngeal carcinoma (Fig. EV3) and, more importantly, the strong yet heterogeneous Trop2 expression in the liver metastasis (Fig. 6G,H). With this solid evidence, the patient successfully entered a clinical trial at our hospital evaluating a Trop2-targeted ADC in solid tumours.

Patient 2 was a postoperative ovarian cancer patient who received [¹⁸F]-FDG PET/CT examination twice for preoperative staging and postoperative surveillance of recurrence and metastasis. On the second [¹⁸F]-FDG PET/CT images, [¹⁸F]-FDG uptake in enlarged right hilar and mediastinal lymph nodes and suspicious bone uptake were reported (Fig. 7A−C), which were not reported on the preoperative [¹⁸F]-FDG PET/CT examination (Fig. EV4). The patient was subjected to [⁶⁸Ga]Ga-NOTA-T4 immunoPET imaging for differential diagnosis. No conspicuous uptake was noticeable on the [⁶⁸Ga]Ga-NOTA-T4 images in these suspicious lesions (Fig. 7D−F). IHC of the biopsied mediastinal lymph node exhibited low and heterogeneous expression of Trop2 with an overall positivity of 30%. Figure 7G shows low Trop2 expression in the biopsied tissue, while Fig. 7H shows negative staining in most of the biopsied tissue.

Patient 3 had a right upper lung mass with multiple systemic metastases involving the liver, bones, and lymph nodes. Figure 8A−C demonstrates increased [¹⁸F]-FDG uptake in both the primary and multiple metastases. However, no significant uptake was observed on [⁶⁸Ga]Ga-NOTA-T4 immunoPET/CT imaging (Fig. 8D−F). Histopathological examination of the biopsied right upper lung mass indicated characteristics of small-cell lung cancer (Fig. EV5). In line

with negative tracer uptake on [⁶⁸Ga]Ga-NOTA-T4 immunoPET images, negative Trop2 expression was further reported (Fig. 8G, H). We also provided the results of strongly positive IHC staining for Trop2 in tissue from a patient with triple-negative breast cancer as a criterion for defining the low-expression level (Appendix Fig. S9). In summary, [⁶⁸Ga]Ga-NOTA-T4 immunoPET can noninvasively visualise the heterogeneous expression of Trop2 in different types of cancers, selecting suitable patients for Trop2-targeted therapies. The correlation between [⁶⁸Ga]Ga-NOTA-T4 uptake value and definite Trop2 expression, as well as the differential diagnostic value of [⁶⁸Ga]Ga-NOTA-T4 immunoPET in inflammatory conditions and cancers/lymph node metastases, are being thoroughly investigated in our ongoing clinical trials.

# Discussion

Existing imaging techniques such as CT or MRI, which are used to diagnose malignant tumours, can indicate the location, size, and anatomical structure of the lesion, but can not reveal alterations at the molecular and cellular levels (Böhmer et al, 2021; Morris and Perkins, 2012). The PET tracer [¹⁸F]-FDG is commonly used in clinical practice to reflects the glucose metabolism of the lesion but is not capable of visualising biomarker expression levels or dynamic changes (Salas and Clark, 2022), and thus is of limited value in optimising molecularly targeted therapies and immunotherapies.

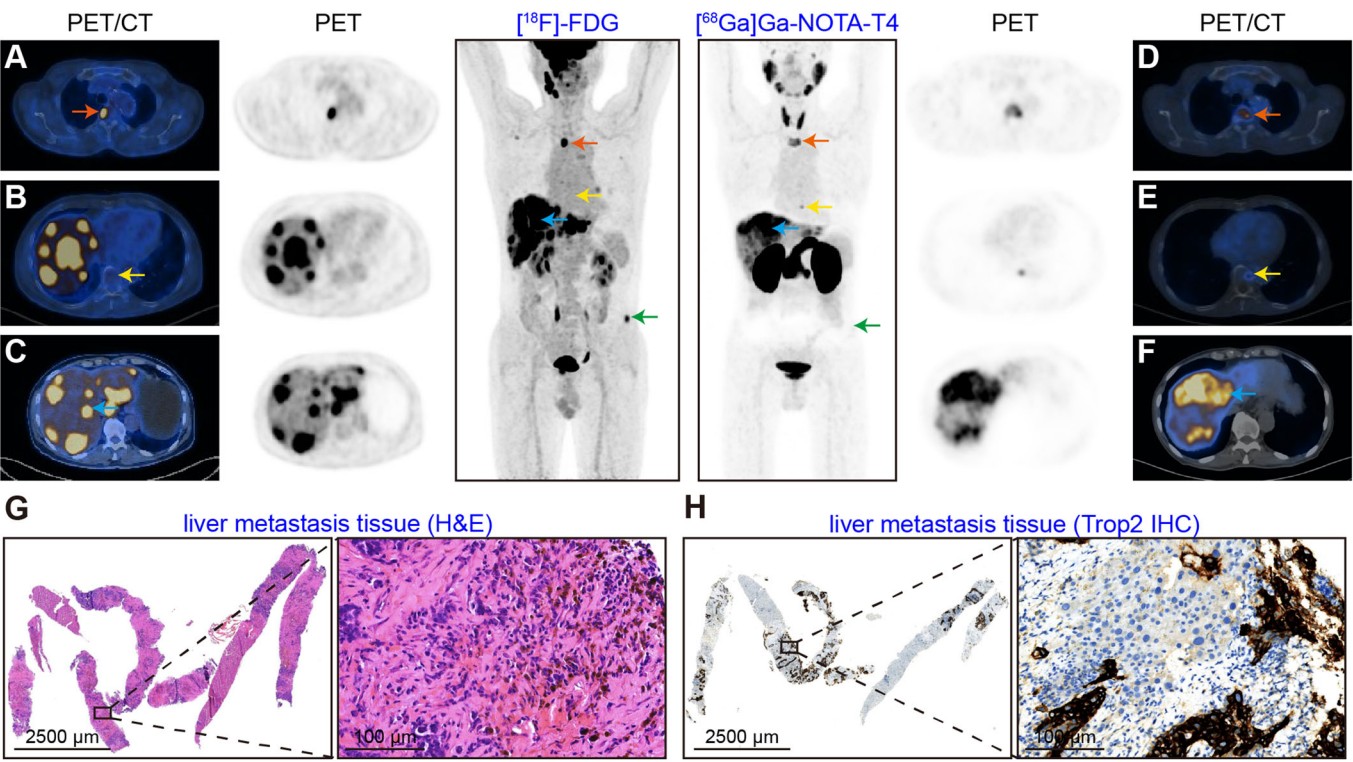

**Figure 6.** [18F]-FDG and [68Ga]Ga-NOTA-T4 PET/CT imaging in a patient with Trop2-positive metastatic nasopharyngeal carcinomas.

A patient diagnosed with nasopharyngeal carcinoma and multiple metastases in the liver, bones, and lymph nodes underwent sequential [18F]-FDG (A−C) and [68Ga]Ga-NOTA-T4 PET/CT (D−F) scans. While the primary tumour and several metastases with intense [18F]-FDG uptake, [68Ga]Ga-NOTA-T4 immunoPET/CT identified bone metastases and multiple liver metastases, indicating positive expression of Trop2 in these metastatic lesions. Notably, a newly formed bone metastatic lesion (yellow arrows) with [68Ga]Ga-NOTA-T4 uptake was identified. H&E (G) and Trop2 IHC (H) staining of the biopsied liver metastasis. Scale bar: 2500 μm and 100 μm. Source data are available online for this figure.

At present, there is a plethora of newly emerging oncological treatment modalities that are rapidly advancing. One must understand the tumour's underlying biology to personalise treatment for each patient's lesion. This could necessitate the use of different molecular biomarkers and assess the extent of targeted drug uptake to predict and monitor the patient's response to therapy. Tumour heterogeneity complicates the identification of the biomolecular characteristics of a patient's malignant tissue and the selection of effective treatments (Burrell et al, 2013; Moek et al, 2017). The primary goal of immunoPET imaging is to stratify patients by detecting the expression of specific biomarkers and to select patients with moderate or high expression of specific biomarkers who may benefit from targeted treatments or immunotherapies (Knowles and Wu, 2012; Wei et al, 2020). Furthermore, immunoPET imaging with radiolabelled therapeutic antibodies enables an objective and comprehensive assessment of receptor expression, pharmacokinetics, and pharmacodynamics of the antibodies. This leads to more scientific and efficient diagnostic and treatment planning for tumour patients. Additionally, it enables monitoring of treatment efficacy and early detection of tumour recurrence, leading to personalised and precise medical treatment (de Vries et al, 2018).

In this study, we developed two nanobody tracers (i.e., [68Ga]Ga-NOTA-T4 and [68Ga]Ga-NOTA-T5) with rapid and strong binding to Trop2 up to the pM binding level. We validated its clearance mode and diagnostic capability in tumour-free nude mice and preclinical pancreatic cancer CDX models with high Trop2 expression. The diagnostic ability of [68Ga]Ga-NOTA-T4 was better than that of [68Ga]Ga-NOTA-T5 in tumour-bearing mice, as supported by higher tumour-to-organ ratios (Appendix Fig. S5). Therefore, [68Ga]Ga-NOTA-T4 is chosen for subsequent immuno-PET imaging studies. We further investigated the impact of co-injection of different unlabelled "cold" T4 doses on radioactivity uptake in major organs, focusing on whether co-injection of unlabelled T4 would result in higher tumour-to-organ ratios, especially tumour-to-kidney ratio. The results demonstrated reduced radioactivity uptake in major organs (e.g., kidney, liver, etc.) with increasing doses of unlabelled T4 and a concomitant decrease in the tumour uptake. With the current data, we found that co-injection of unlabelled T4 did not yield the expected outcome of balanced high tumour uptake and low kidney accumulation. This suggested that other engineering strategies, including polyethylene glycol modification (Lee et al, 2021) and removal of His-tag (D'Huyvetter et al, 2014), are needed to reduce kidney accumulation and reduce nephrotoxicity when developing α/β emitter-labelled T4 for radioimmunotherapy (Wei et al, 2022; Yang et al, 2022).

Building on our previous research (Liang et al, 2023; Zhang et al, 2023), we developed a nanobody derivative (ABDT4, 21 kDa) from T4 that is capable of simultaneously binding to Trop2 and

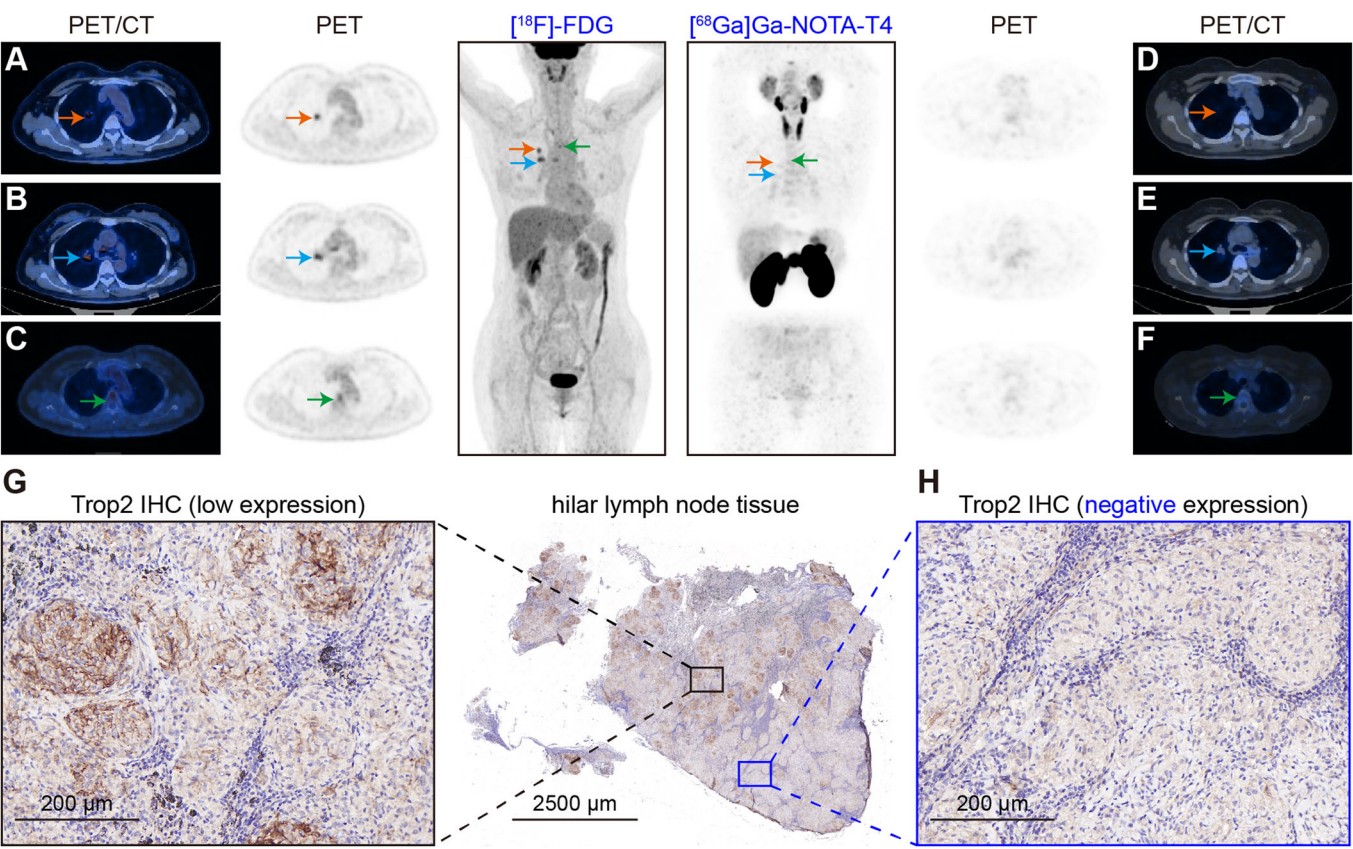

**Figure 7. [¹⁸F]-FDG and [⁶⁸Ga]Ga-NOTA-T4 PET/CT imaging in a patient with suspicious lung cancer.**

A patient with postoperative ovarian cancer with enlarged right hilar and mediastinal lymph nodes underwent [¹⁸F]-FDG (A−C) and [⁶⁸Ga]Ga-NOTA-T4 (D−F) PET/CT examinations. Suspicious lesions/lymph nodes in the hilum of the right lung and mediastinum, as well as a suspicious bone uptake, were found on [¹⁸F]-FDG PET/CT images. In contrast, no abnormal [⁶⁸Ga]Ga-NOTA-T4 uptake was seen in these suspicious lesions. (G, H) IHC staining of the biopsied right hilar lymph node tissue showed low Trop2 expression with a positive rate of 30%. The left panel (G) showed a low expression of Trop2, and the right panel (H) presented a negative expression. Scale bar: 2500 μm and 200 μm. Source data are available online for this figure.

human/murine serum albumin. Administration of ABDT4 into mice 48 h in advance allowed sufficient saturation of Trop2 antigens on the surface of tumour cells due to its prolonged circulation time. As expected, [⁶⁸Ga]Ga-NOTA-T4 immunoPET imaging in the ABDT4-blocking group yielded significantly lower tumour uptake when compared to that in the non-blocking group, confirming the tracer's specificity towards Trop2. Besides, ABDT4 is feasible for labelling therapeutic radionuclides to achieve radioimmunotherapy due to its long circulation time in vivo. The total-body PET/CT imaging technique guides the development of radiopharmaceuticals by visualising the real-time dynamic pharmacokinetics (Cherry et al, 2017). In preclinical experiments involving a beagle dog, one hour of dynamic imaging was performed after a bolusinjection of [⁶⁸Ga]Ga-NOTA-T4. The results reaffirmed that the tracer was excreted through the urinary system. There was minimal radioactivity uptake in all other tissues/organs, including the bones, indicating excellent pharmacokinetics and stability of the tracer.

Following thorough preclinical studies, we carried out a pilot clinical trial in which ten subjects were recruited to undergo [⁶⁸Ga]Ga-NOTA-T4 scans, three of whom underwent sequential [¹⁸F]-FDG and [⁶⁸Ga]Ga-NOTA-T4 scans. Kidneys, pancreases, and glands (thyroid

and salivary glands) showed obvious [⁶⁸Ga]Ga-NOTA-T4 uptake, whereas the uptake in other major organs and tissues was low, the result that coincided with previous studies (Stepan et al, 2011; Tsujikawa et al, 1999). Overall, [⁶⁸Ga]Ga-NOTA-T4 has a cleaner imaging background than [¹⁸F]-FDG. In one patient with nasopharyngeal carcinoma, [⁶⁸Ga]Ga-NOTA-T4 immunoPET/CT precisely identified multiple metastases and strong Trop2 expression in liver metastasis, which was confirmed by IHC staining. Based on the valuable information, the patient entered into Trop2-targeted ADC treatment. In the second patient with ovary cancer surgery history and suspicious [¹⁸F]-FDG−avid lesions/lymph nodes on regular follow-up, [⁶⁸Ga]Ga-NOTA-T4 showed no evident uptake in these lesions and the subsequent biopsy confirmed inflammatory lymph nodes. The tracer's specificity in imaging Trop2 expression was further confirmed in a third patient with Trop2-negative small-cell lung cancer. A preclinical study found that sacituzumab govitecan (SG), a Trop2-targeted ADC, was efficacious in cases with higher Trop2 expression. However, it also exhibited growth inhibitory effects in tumours displaying low to moderate Trop2 expression (Zhao et al, 2023). In addition, in a clinical trial examining targeted ADC therapy for Trop2 in triple-negative breast cancer, patients who showed higher Trop2 expression displayed a tendency for elevated objective response rates and longer

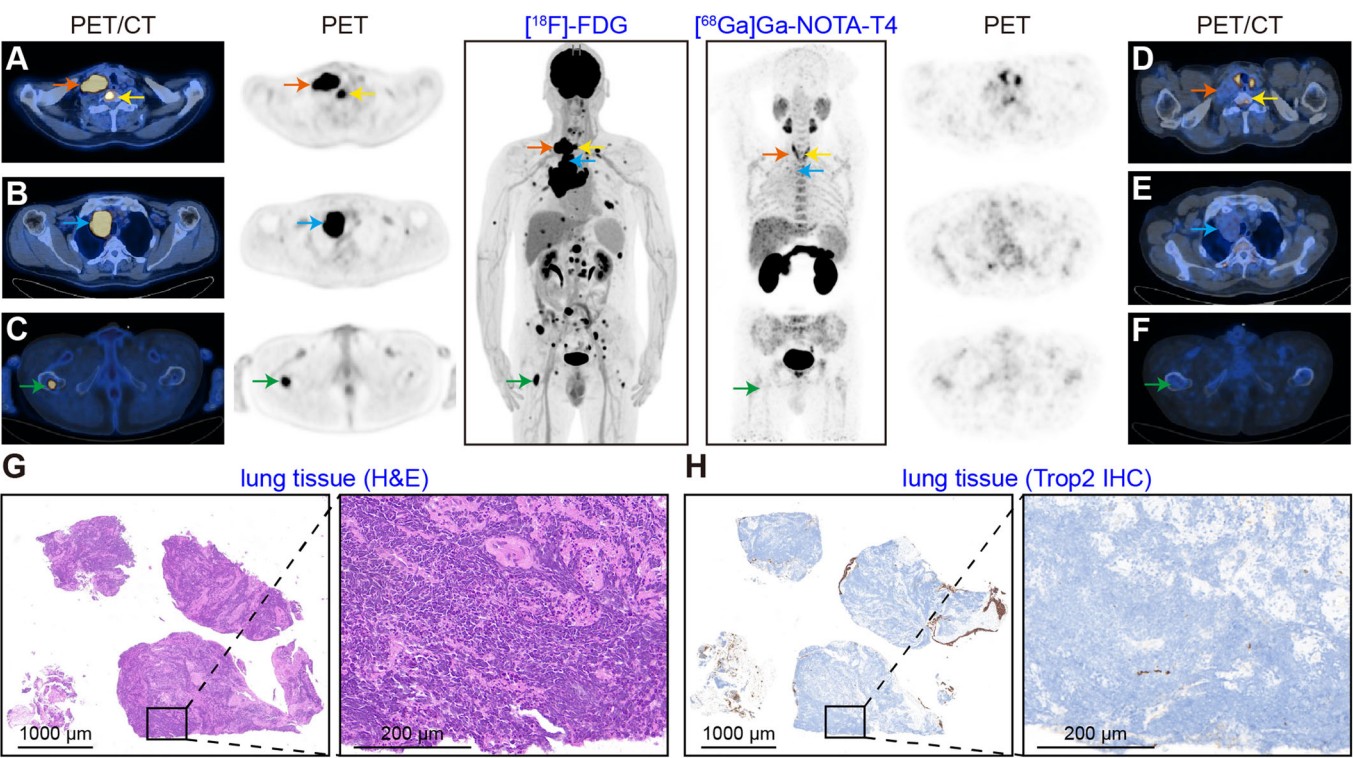

**Figure 8. [¹⁸F]-FDG and [⁶⁸Ga]Ga-NOTA-T4 PET/CT imaging in a patient with small-cell lung cancer.**

A patient with small-cell lung cancer with multiple systemic metastases underwent sequential [¹⁸F]-FDG (A−C) and [⁶⁸Ga]Ga-NOTA-T4 (D−F) PET/CT within a week. The four lesions, including primary and metastatic lesions labelled, were noted with high [¹⁸F]-FDG uptake but did not show significant [⁶⁸Ga]Ga-NOTA-T4 uptake. H&E (G) and Trop2 IHC (H) staining of the biopsied right lung mass. IHC staining result shows negative Trop2 expression. Scale bar: 1000 μm and 200 μm. Source data are available online for this figure.

progression-free survival in comparison to those with lower expression level (Bardia et al, 2021b). Thus, [⁶⁸Ga]Ga-NOTA-T4 has the potential to visualise heterogeneous expression and dynamics of Trop2 noninvasively, improving clinical decision-making when considering Trop2-targeted therapeutics as treatment options.

In recent years, rapid development has occurred with ADCs in oncology therapy. ADCs, consisting of antibodies linked to cytotoxic drugs through chemical links, exhibit higher anti-tumour potency and a broader therapeutic window than traditional chemotherapeutic agents for tumours (Chau et al, 2019). Nevertheless, ADC therapy targeting Trop2 is also subject to toxicities due to Trop2 expression in normal tissues. For instance, PF-06664178, a Trop2-targeted ADC bearing Aur0101 (a novel inhibitor of microtubule proteins), caused skin rash and mucosal inflammation as dose-limiting toxicities in a Phase I study of adult patients with advanced solid tumours (King et al, 2018). In addition, a preclinical study performed in nonhuman primates, specifically the cynomolgus monkeys, reported toxicity signals in the oral mucosa and oesophagus, among others (Strop et al, 2016). Another ADC targeting Trop2, Datopotamab deruxtecan (Dato-potamab-DXd), was found to cause predominantly non-haematologic adverse events in 33% of patients, with grade 3 or higher events occurring most commonly as stomatitis (13%) (Tarantino et al, 2021). SHR-A1921 is a novel ADC composed of a humanised anti-Trop2 IgG1 mAb attached to DNA

topoisomerase I inhibitor via a tetrapeptide-based cleavable linker. Despite the good objective response rate of 33.3% and disease control rate of 80.0% in patients with advanced solid tumours, a recent first-in-human study reported that the drug is associated with treatment-related adverse events (TRAEs), including nausea, stomatitis, anaemia, vomiting, decreased appetite, decreased weight, and rash (Wang et al, 2023). The most common Grade ≥3 TRAE was stomatitis, which may be partially associated with intensive accumulation of SHR-A1921 in the salivary glands and controlled drug release into the mouth.

Stepan et al discovered Trop2 expression in tissues, including the cervix, oesophagus, skin, salivary glands, mammary glands, kidneys, pancreas, and prostate. However, it was absent in tissues such as the brain, bone marrow, colon, heart, muscle, ovary, spleen, and thyroid (Stepan et al, 2011). In addition to radioactivity uptake in the salivary glands, pancreas, and kidneys, radioactivity uptake in the thyroid was also observed on [⁶⁸Ga]Ga-NOTA-T4 images of the three subjects in this study, which is different from what has been previously reported in the literature. It is reasonable to assume that the physiological uptake of Trop2-targeted tracer in these normal organs/tissues may predict the side effects of Trop2-targeted therapies. Off-tumour on-target toxicity could be predicted not only by observing the radioactivity uptake of [⁶⁸Ga]Ga-NOTA-T4 in the subject's organs but also by noninvasively visualising the pharmacokinetics and pharmacodynamics of mAbs and ADCs

labelled with long half-life radionuclides ($^{89}$Zr, $^{64}$Cu, etc.), as a way of minimising the risk of possible off-tumour toxicity in normal tissues/organs expressing Trop2. We assume that clinical trial enrollment criteria for Trop2-targeted ADC therapy were not based exclusively on Trop2-positive expression and thus did not result in highly satisfactory treatment efficacy (Bardia et al, 2021a). Moreover, a clinical trial evaluating [$^{89}$Zr]Zr-DFO-SHR-1920 (a naïve mAb of SHR-A1921) is undergoing at our hospital. With [$^{68}$Ga]Ga-NOTA-T4 immunoPET readily available for clinical use, we can noninvasively detect Trop2 expression to assist in clinical decision-making in terms of patient stratification (select Trop2-positive patients for treatment and prevent futile treatment of Trop2-negative patients) and response monitoring (assess the response of tumours and off-tumour on-target toxicity).

In summary, we developed two nanobody radiopharmaceuticals targeting human Trop2 with one of the tracers ([$^{68}$Ga]Ga-NOTA-T4) showing favourable pharmacokinetics and promising diagnostic capabilities in both preclinical studies and a pilot clinical trial. [$^{68}$Ga]Ga-NOTA-T4 immunoPET could visualise the heterogeneous expression of Trop2 to screen patients that might benefit from Trop2-targeted therapies and evaluate the responses after administration of therapeutics, hopefully optimising clinical management of specific types of solid tumours.

# Methods

## Production and binding affinity test of Trop2 nanobodies

The Trop2-Fc fusion protein used for immunisation has the following sequence: MDMRVPAQLLGLLLLWFPGSRCHTAAQD NCTCPTNKMTVCSPDGPGGRCQCRALGSGMAVDCSTLTSKC LLLKARMSAPKNARTLVRPSEHALVDNDGLYDPDCDPEGRFK ARQCNQTSVCWCVNSVGVRRTDKGDLSLRCDELVRTHHILID LRHRPTAGAFNHSDLDAELRRLFRERYRLHPKFVAAVHYEQPT IQIELRQNTSQKAAGDVDIGDAAYYFERDIKGESLFQGRGGLD LRVRGEPLQVERTLIYYLDEIPPKFSMKRLTEPKSCDKTHTCPP CPAPELLGGPSVFLFPPKPKDTLMISRTPEVTCVVVDVSHEDPE VKFNWYVDGVEVHNAKTKPREEQYNSTYRVVSVLTVLHQD WLNGKEYKCKVSNKALPAPIEKTISKAKGQPREPQVYTLPPSR EEMTKNQVSLTCLVKGFYPSDIAVEWESNGQPENNYKTTPPV LDSDGSFFLYSKLTVDKSRWQQGNVFSCSVMHEALHNHYTQK SLSLSPGK. Trop2-Fc was recombinantly produced in HEK293 cells and used to immunise a camel five times. Then, nanobodies specific for Trop2 were produced following standard experimental procedures, including library construction, phage display, and recombinant expression in Chinese hamster ovary (CHO) cells under strict conditions meeting translational requirements. The endotoxin of the nanobodies was below 1EU/mg. The expression yield of two selective His-tagged nanobodies (i.e., T4 and T5) was 166.00 mg/L and 170.35 mg/L in CHO cells, respectively. The binding kinetics between the two nanobodies and the immobilised human Trop2 protein (TR2-H5223, ACRO Biosystems) was verified using the surface plasmon resonance (SPR) interaction test. ABDT4 was measured for binding affinity to recombinant human Trop2 protein, human serum albumin, and murine serum albumin. The equilibrium dissociation degree (M), denoted by the $K_D$, signified the ultimate binding affinity. This value was obtained by dividing the dissociation rate ($k$d) by the association rate constant ($k$a) ($M^{-1}\,s^{-1}$).

## Cell culture and cell-derived xenograft (CDX) mouse models

The T3M-4 pancreatic cancer cell line was obtained from MeisenCTCC (Zhejiang, China). The cell line is authenticated and tested for mycoplasma contamination every three months. Cells were cultured in RPMI 1640 medium (Shanghai BasalMedia Technologies Co., Ltd.) supplemented with 10% foetal bovine serum (FBS; GE Healthcare, Chicago, IL, USA) and 1% penicillin/streptomycin (Invitrogen). The cell line was cultured in an incubator at 37 °C with 5% $CO_2$ humidity. All experimental protocols and animal care procedures followed the principles of the Institutional Committee for the Care and Use of Animals (Renji Hospital, Shanghai Jiao Tong University School of Medicine). Mice are housed under specified pathogen-free conditions at Renji Hospital. Sterile phosphate-buffered saline (PBS, HyClone) resuspension of $5 \times 10^6$ cells followed by mixing the cell suspension with an equal volume of matrigel matrix (Corning), which was used to construct a subcutaneous CDX nude mouse (4–6 weeks female nude mouse, GemPharmatech) model. The cell suspension was injected into the left side of the mice's abdomen. Mouse models bearing tumours were performed for in vivo immunoPET imaging when the diameter was next to 6–10 mm around 2 weeks.

## Preparation and quality control of nanobody-derived tracers

The labelling precursor NOTA-T4/T5 was prepared the same way as in the previous study (Huang et al, 2024). Briefly, 2 mg of nanobody (T4 or T5) was dissolved in sterile PBS (pH 7.4) and pH was adjusted to 9–10 with 0.5 M $Na_2CO_3$. The nanobody solution was mixed with the bifunctional chelator 2-S-(4-Isothiocyanato-benzyl)-1,4,7-triazacyclononane-1,4,7-triacetic acid (*p*-SCN-Bn-NOTA; Macrocyclics) fully dissolved in dimethyl sulfoxide (DMSO) at a molar ration of 1:10. The mixture was shaken for two hours at room temperature. The product of the reaction was purified and concentrated using equilibrated PD-10 desalting columns (GE Healthcare) and Amicon Ultra Centrifugal Filters (10 kDa, Merck) and then stored at 4 °C for subsequent labelling experiments in three months. For radiolabelling, about 27 nmol of NOTA-nanobody was put into one mCi of $^{68}$Ga eluent with the pH adjusted to 4–4.5, and the reaction went for 5–10 min at room temperature, followed by purification of the final product with equilibrated PD-10 column. Generally, high-purity product [$^{68}$Ga]Ga-NOTA-T4 or [$^{68}$Ga]Ga-NOTA-T5 was obtained after purification. The instant thin-layer chromatography (iTLC; Eckert & Ziegler Radiopharma Inc.) detected the radiochemical purity of [$^{68}$Ga]Ga-NOTA-T4 and [$^{68}$Ga]Ga-NOTA-T5. To prepare for preclinical and clinical imaging, each [$^{68}$Ga]Ga-NOTA-T4 was filtered through a 0.22-μm Millipore filter to ensure sterility.

## Preclinical immunoPET imaging and data analysis

For all the preclinical studies, both tumour-free and tumour-bearing mice were randomly divided into different imaging groups. After [$^{68}$Ga]Ga-NOTA-T4 or [$^{68}$Ga]Ga-NOTA-T5 was injected into tumour-free mice or CDX model mice for 45 min, the mice under anaesthesia were placed in the prone position on the microPET/CT bed to perform the image acquisition. For the ABDT4-blocking

## The paper explained

### Problem

Diagnostic imaging methods for malignant tumours, such as computed tomography (CT) and magnetic resonance imaging (MRI), are currently unable to capture changes at the molecular or cellular levels. Invasive histological biopsy faces inherent limitations and detection biases due to the heterogeneous nature of the tumours. Thus, noninvasive techniques are urgently needed to visualise the heterogeneous expression of biomarkers such as trophoblast cell surface antigen 2 (Trop2), a promising oncogenic pan-cancer biomarker.

### Results

Two novel nanobody tracers (i.e., [68Ga]Ga-NOTA-T4 and [68Ga]Ga-NOTA-T5) with high binding affinity were developed to specifically target Trop2. Preclinical studies in tumour-bearing mouse models revealed these tracers' good pharmacokinetics and diagnostic potentials. [68Ga]Ga-NOTA-T4 showed excellent pharmacokinetics in a beagle dog, which warrants translational potential. In three patients with solid tumours, [68Ga]Ga-NOTA-T4 immuno-positron emission tomography (immunoPET) noninvasively and precisely visualised negative, positive, and heterogeneous Trop2 expression in primary and metastatic tumours, which was confirmed by immunohistochemistry staining.

### Impact

The two novel Trop2-targeted nanobody tracers exhibit promising diagnostic value and could allow the identification of patients who would benefit from Trop2-targeted therapies and the evaluation of post-treatment therapeutic responses.

group, unlabelled ABDT4 was injected into mice 48 h in advance (400 μg per mouse) to fully bind to the target on the surface of the tumour cells to saturate the targets. In addition, different doses of unlabelled T4 (50 μg, 200 μg, 400 μg per mouse) were injected into the mice to investigate the changes in radioactivity uptake of the major organs. An IRIS PET/CT system (Inviscan Imaging Systems) was used to acquire microPET data successively. The nonscatter-corrected 3D-ordered subset expectation optimisation/maximum a posteriori (OSEM3D/MAP) approach was used to recover the PET data. OsiriX Lite (Pixmeo SARL) and Inveon Research Workplace (Siemens Preclinical Solutions) were used to evaluate the data. Upon completion of the scan, the radioactivity uptake of the major organs/tissues (heart, lung, liver, kidney, and muscle) was quantified by outlining the region of interest (ROI), presenting values in %ID/g.

## Preclinical Beagle dog dynamic data acquisition on total-body PEC/CT scanner

An adult male beagle dog (weighing ~12.5 kg and 1.5 years old) was anaesthetised using an anaesthesia mixture (Zoletil and Xylazine). The anaesthetised beagle dog was supine for total-body PET/CT (uEXPLORER, United Imaging Healthcare) at Renji Hospital after approval following the approved imaging protocol. After the [68Ga]Ga-NOTA-T4 (almost 111 MBq) was injected into a vein in the hind arm of the beagle dog, a 60-minute dynamic scan was acquired. The dynamic sequence of 92 frames of images was quantitatively analysed using Pmod4.0 software (PMOD 4.0 Fusion, PMOD Technologies LLC, Zurich, Switzerland). Whole-body radioactivity biodistributions were calculated as TACs

describing the dynamic uptake patterns in the major organs of the beagle dog.

## Biodistribution and staining studies after preclinical immunoPET imaging

At the end of the imaging, mice were euthanised. The distribution of radioactivity in the tumour and the major organs/tissues (blood, skin, muscle, bone, heart, lung, liver, kidney, pancreas, spleen, stomach, intestine, and brain) was examined at an automated gamma counter (PerkinElmer, WIZARD² 2470). Tumour tissues fixed in paraformaldehyde tissue fixative were scheduled for subsequent H&E staining and IHC staining using a commercial anti-Trop2 antibody (sc-376746, Santa Cruz Biotechnology). Two dilution ratios (1:50 and 1:100) were tested, and the dilution ratio 1:50 was chosen in our studies.

## Pilot clinical PET/CT imaging in patients with solid tumours

Ethical approval for the clinical trial in the study was obtained from the Institutional Review Board of Huashan Hospital, Fudan University (2023-1017). The study was registered as a prospective clinical trial (ClinicalTrials.gov Identifier: NCT06203574). The clinical imaging studies conformed to the WMA Declaration of Helsinki principles and the Department of Health and Human Services Belmont Report. The inclusion criteria were as follows: Being between 18 and 80 years of age and of either sex; diagnosed with solid tumours confirmed by biopsy or surgical pathology; imaging revealed suspicious lymph nodes or distant metastases; informed consent must be signed in writing by the subject or their legal guardian or caregiver; willingness and ability to cooperate with all programmes of this study. The exclusion criteria were as follows: severe hepatic and renal insufficiency; renal function: serum creatinine more than or equal to the upper limit of the normal range; liver function: bilirubin, glutamic pyruvic transaminase or glutamic oxaloacetic transaminase more than or equal to the upper limit of the normal range. As this is a diagnostic clinical trial, all participants were informed about the trial and signed a consent form. Since it is not a blinded trial, participants were aware of their group assignment. No additional special preparation is required for [68Ga]Ga-NOTA-T4 PET/CT examination. Patients were administered a [68Ga]Ga-NOTA-T4 dose at $130.61 \pm 36.04$ MBq. Adverse events were noted up to 4 hours after the injection of [68Ga]Ga-NOTA-T4. Forty-five minutes after injection, static PET/CT imaging was performed using a PET/CT scanner (uMI510, United Imaging, Shanghai, China). [68Ga]Ga-NOTA-T4 PET/CT scans were conducted from the head region to the upper thigh. Three patients underwent traditional [18F]-FDG PET/CT examination before [68Ga]Ga-NOTA-T4 PET/CT scanning. The acquired data were transferred to a dedicated workstation (United Imaging, Shanghai, China) for analysis. Image reconstruction was performed using a standard ordered subset expectation maximisation (OSEM) algorithm. For quantitative assessment, the maximum standardised uptake value ($SUV_{max}$) was utilised to quantify the uptake of radiopharmaceuticals by tumour primary and metastatic lesions.

## Statistical analysis

Statistical analysis was performed using Prism software (Version 10.0, GraphPad Software) and presented as means ± standard

deviation (SD). Significance was determined using the two-tailed unpaired $t$ test, two-way ANOVA. The two groups' differences are considered significant when $*P < 0.05$ and $**P < 0.01$ are present.

## Data availability

This study includes no data deposited in external repositories.

## Peer review information

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

## Acknowledgements

The work was supported in part by the National Key Research and
Development Program of China (Grant No. 2021YFA0910000), the National
Natural Science Foundation of China (Grant No. 82372014), and the Shen
Kang-United Imaging Joint Research and Development Plan (Grant No.
SKLY2022CRT301).

## Author contributions

**Wei Huang**: Conceptualisation; Data curation; Formal analysis; Investigation;
Methodology; Writing—original draft. **You Zhang**: Data curation; Formal
analysis; Investigation; Methodology; Writing—original draft. **Min Cao**: Data
curation; Formal analysis; Investigation; Methodology; Writing—original draft.
**Yanfei Wu**: Data curation; Investigation; Methodology. **Feng Jiao**: Investigation;
Methodology. **Zhaohui Chu**: Resources; Investigation; Methodology. **Xinyuan
Zhou**: Investigation; Methodology. **Lianghua Li**: Data curation; Investigation;
Methodology. **Dongsheng Xu**: Investigation; Methodology. **Xinbing Pan**:
Investigation; Methodology. **Yihui Guan**: Resources; Methodology; Writing—
review and editing. **Gang Huang**: Resources; Funding acquisition; Visualisation;
Writing—review and editing. **Jianjun Liu**: Conceptualisation; Resources;
Supervision; Project administration; Writing—review and editing. **Fang Xie**:
Conceptualisation; Resources; Supervision; Investigation; Writing—review and
editing. **Weijun Wei**: Conceptualisation; Resources; Formal analysis;
Supervision; Funding acquisition; Investigation; Methodology; Writing—original
draft; Project administration; Writing—review and editing.

## Disclosure and competing interests statement

W Wei, W Huang and J Liu are co-inventors on a provisional patent application
encompassing the technology reported in the manuscript. W Wei is a
consultant of Alpha Nuclide (Ningbo) Medical Technology Co., Ltd. The
remaining authors declare no competing interests.

# Expanded View Figures

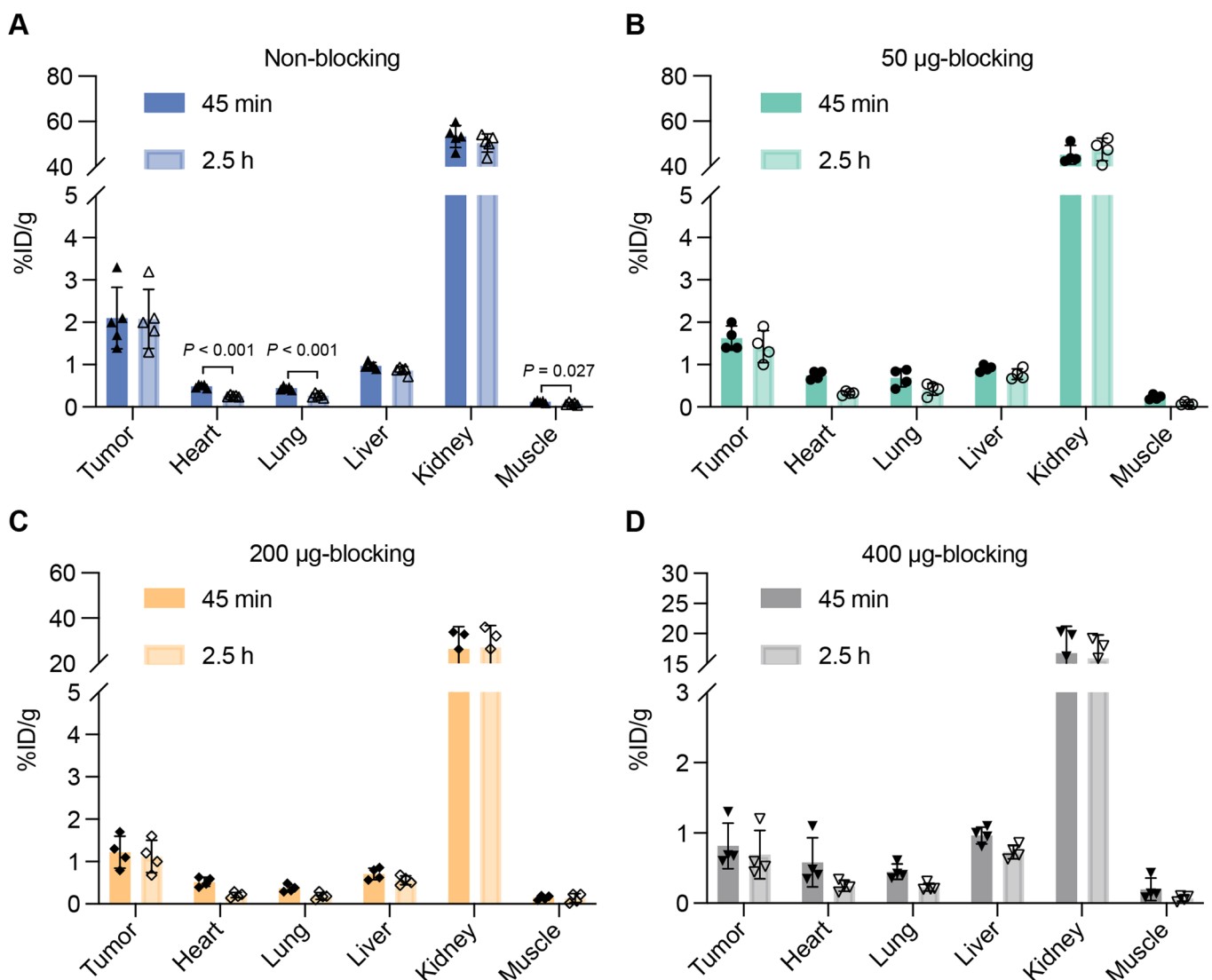

**Figure EV1.  [⁶⁸Ga]Ga-NOTA-T4 uptake in the non-blocking and three blocking groups.**

Comparison of radioactivity uptake in tumour and major organs at two time points (45 min and 2.5 h) in the non-blocking group (**A**, $n = 5$), 50 μg-blocking (**B**, $n = 6$), 200 μg-blocking (**C**, $n = 6$), and 400 μg-blocking groups (**D**, $n = 6$). *t* test, mean ratio ± SD.

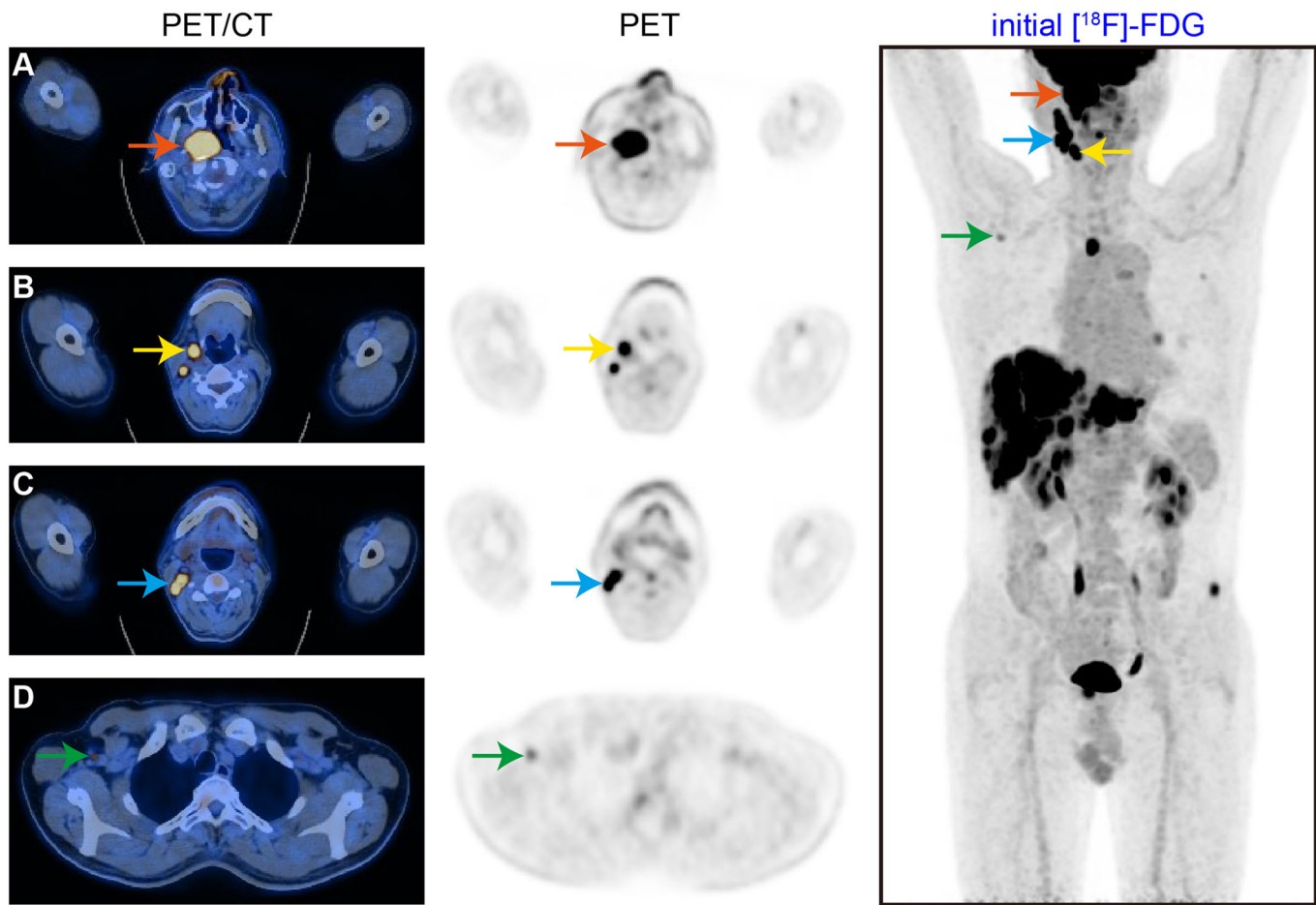

**Figure EV2. The initial baseline [18F]-FDG PET/CT examination of patient 1 with nasopharyngeal carcinoma.**

The primary tumour (**A**, orange arrows) and multiple metastases, including the right parapharyngeal space lymph node (**B**, yellow arrows), right neck lymph node (**C**, blue arrows), bone and liver metastases (Fig. 6A–C in the main manuscript), and a suspicious right axillary lymph node metastasis (**D**, green arrows) were presented. Note: the [18F]-FDG MIP image here is the same MIP image in Fig. 6.

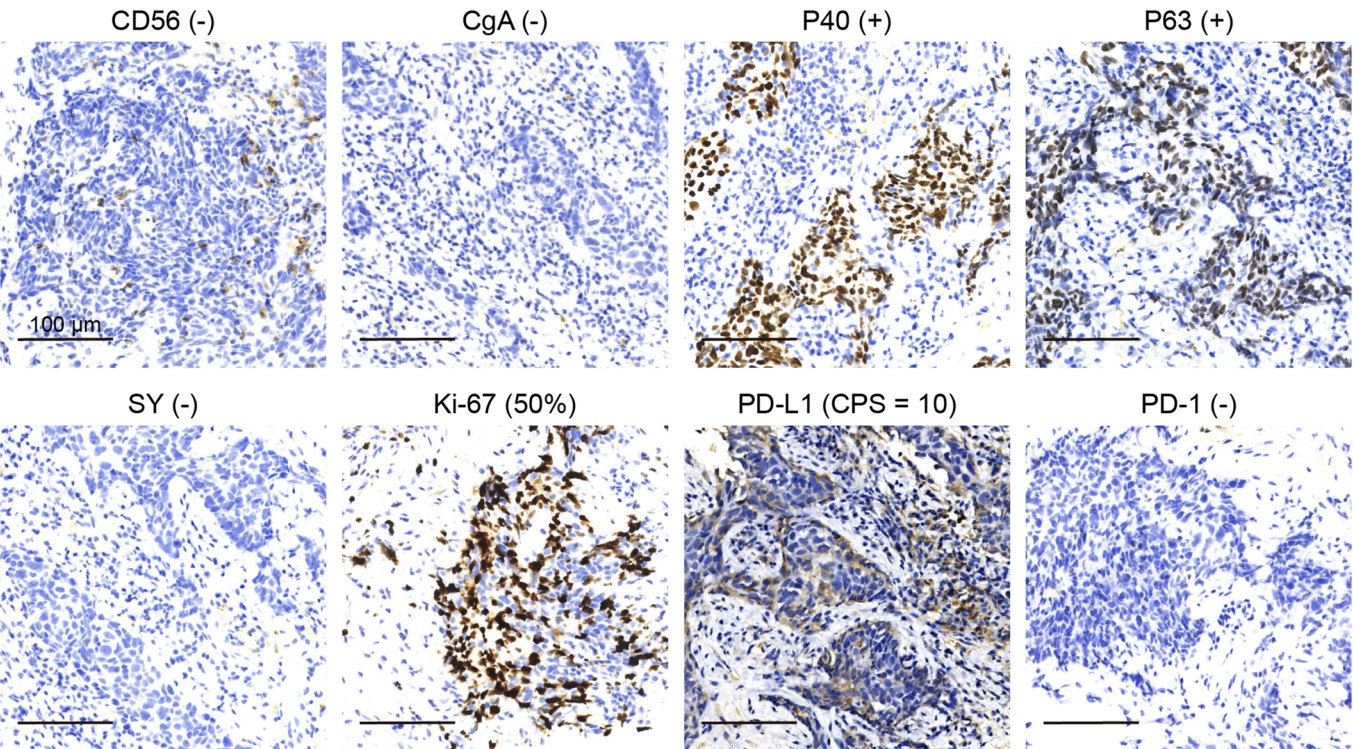

**Figure EV3.  Histopathological examination of the biopsied liver nodule.**

The tested biomarkers confirmed the liver metastasis from poorly differentiated nasopharyngeal carcinoma. "+" means positive expression, and "–" means negative expression. Scale bar: 100 μm.

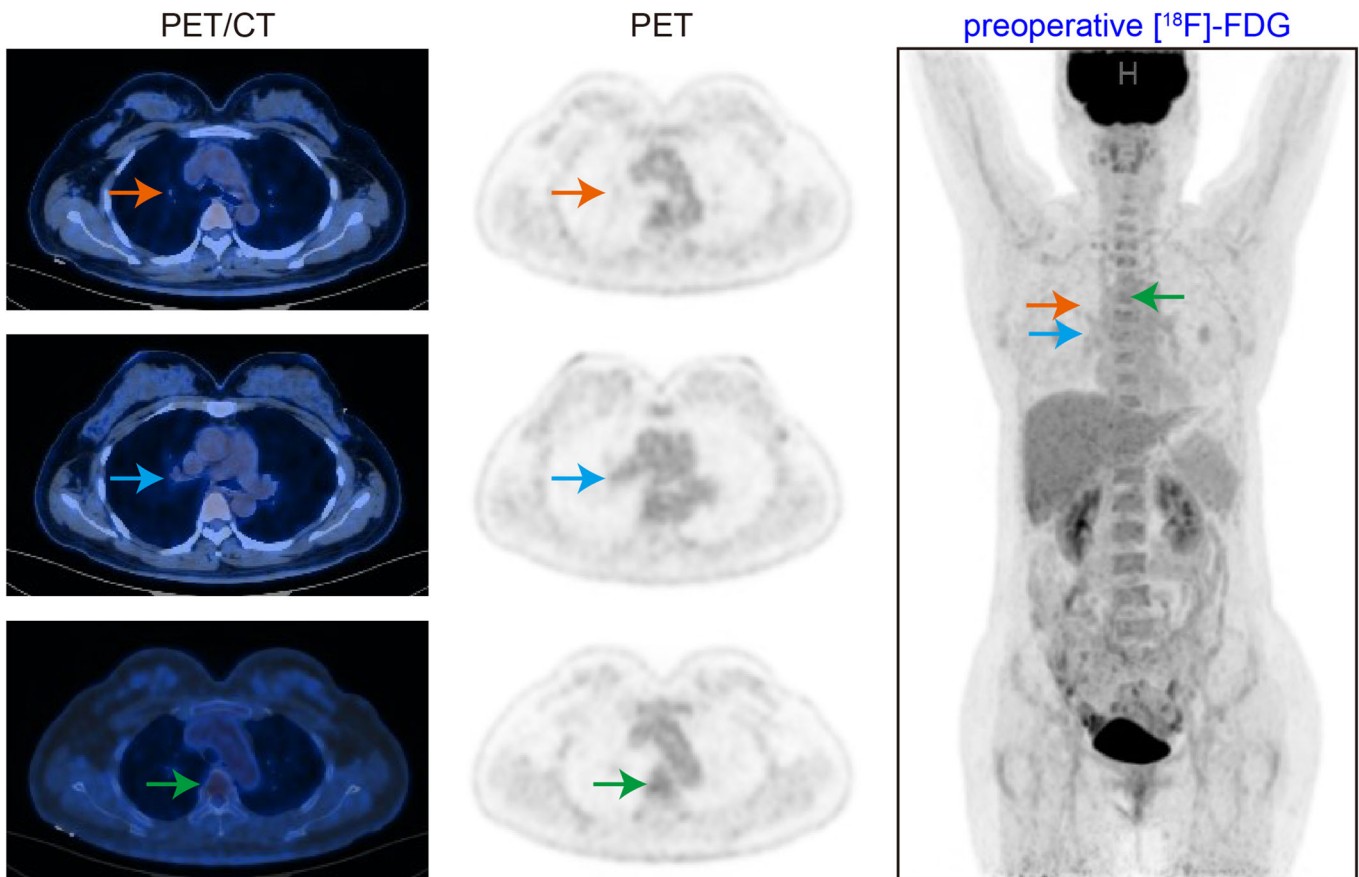

**Figure EV4.  The preoperative [18F]-FDG PET/CT examination of patient 2.**

The lesions and metastasis with enlarged uptake detected on the second [18F]-FDG PET/CT were absent on the first preoperative [18F]-FDG PET/CT. Source data are available online for this figure.

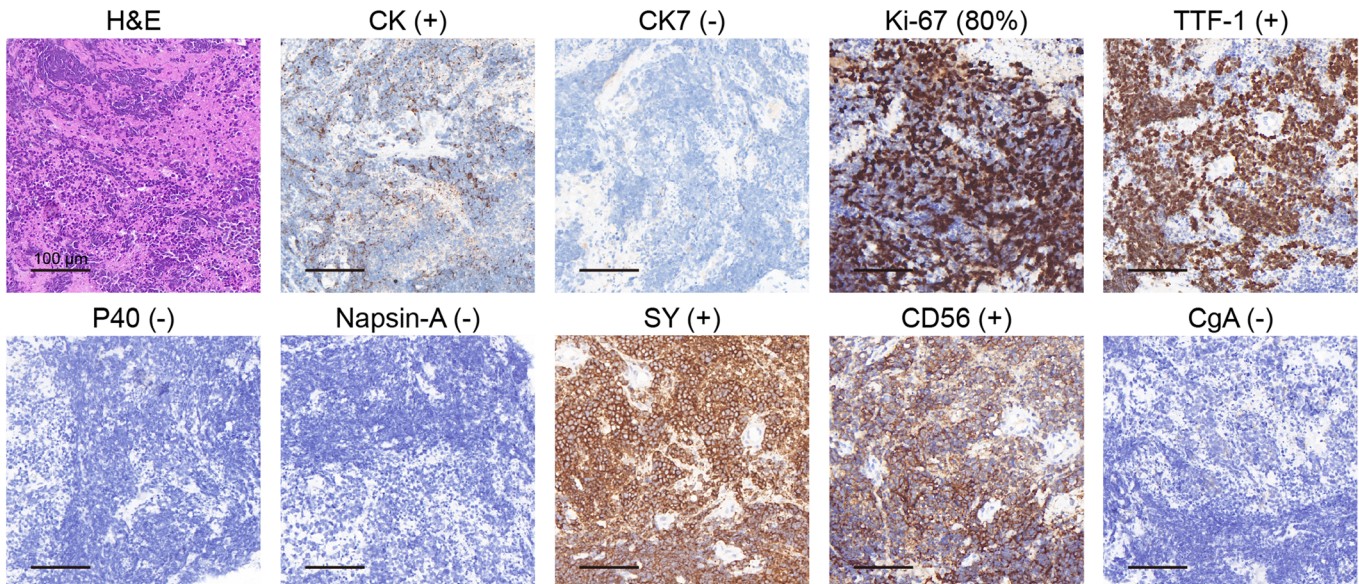

**Figure EV5.   Histopathological examination of the biopsied right upper lung mass.**

The tested biomarkers indicated characteristics of small-cell lung cancer. "+" means positive expression, and "–" means negative expression. Scale bar: 100 μm.

