## [Peer Review File · EMBO Molecular Medicine]

ImmunoPET imaging of Trop2 in patients with solid tumours

Wei Huang, You Zhang, Min Cao, Yanfei Wu, Feng Jiao, Zhaohui Chu, Xinyuan Zhou, Lianghua Li, Dongsheng Xu, Xinbing Pan, Yihui Guan, Gang Huang, Jianjun Liu, Fang Xie, and Weijun Wei

Corresponding authors: Weijun Wei (wwei@shsmu.edu.cn) , Fang Xie (fangxie@fudan.edu.cn), Jianjun Liu (ljsh@sytu.edu.cn)

Review Timeline:

Submission Date:	31st Dec 23
Editorial Decision:	23rd Jan 24
Revision Received:	5th Feb 24
Editorial Decision:	28th Feb 24
Revision Received:	6th Mar 24
Editorial Decision:	7th Mar 24
Revision Received:	8th Mar 24
Accepted:	12th Mar 24

Editor: Lise Roth

Transaction Report:

23rd Jan 2024

Dear Prof. Wei,

Thank you for the submission of your manuscript to EMBO Molecular Medicine. We have now received feedback from the three reviewers who agreed to evaluate your manuscript.

As you will see from the enclosed reports, the referees acknowledge the potential interest of the findings, however they also raise several major concerns, regarding the unclear section on the beagle dog and species specificity, the (lack of) ethic approval and clinical trial registry for study in patients, the tracer production, and overall text clarity, rationale and discussion.

Based on the nature of the concerns and considering that at EMBO Press we encourage a single round of revisions in a limited time frame, we prefer to return the manuscript to you at this point with the decision that we cannot offer to publish it.

Given the potential interest of the findings, we would, however, be willing to consider a new manuscript on the same topic if at some time in the near future you obtained data that would considerably strengthen the message of the study and address the referees concerns in full. To be completely clear, however, I would like to stress that if you were to send a new manuscript this would be treated as a new submission rather than a revision and would be reviewed afresh, in particular with respect to the literature and the novelty of your findings at the time of resubmission. If you decide to follow this route, please make sure you nevertheless upload a letter of response to the referees' comments.

I am sorry that I could not bring better news this time and hope that the referee comments are helpful in your continued work in this area.

Yours sincerely,

Lise Roth

**** Reviewer's comments ****

Referee #1 (Comments on Novelty/Model System for Author):

The authors investigated [68Ga]Ga-NOTA-T4 immunoPET, an Immuno-PET radiotracer to determine Trophoblast cell surface antigen 2 (Trop2) expression in-vivo. This cell surface antigen serves as a potential pan-cancer biomarker in varying tumor entities. In a translational setting, they reported on heterogenous expression of Trop2 to identify individuals that may benefit from Trop2-directed therapies. In this regard, the authors used rodents and could demonstrate successful blocking of radiotracer uptake using ABDT4, indicative for specificity. Experiments in a beagle dog provided favorable biodistribution properties with rapid clearance from the blood pool and other organs such as the liver and expected renal excretion. Those experiments were then followed by patients in a head-to-head-comparison with 18F-FDG (reference), most likely to identify tumor heterogeneity.

While the herein used approach of investigating mice, followed by large animals and finally humans is promising, the results in large animals are confusing, as the Figure Legends in Figure 4 may be not appropriately labeled. It says "Figure 4. ABDT4 blocked tumor uptake of [68Ga]Ga-NOTA-T4 and the dynamic distribution patterns of [68Ga]Ga-NOTA-T4 in a beagle dog.", but Fig. 4A-B may be mice. This should be clarified. While this radiotracer appears promising in murine models, the uptake in humans appears to be inferior to the reference FDG. As such, the authors should further discuss the value of this radiotracer, e.g., depicting tumor heterogeneity in a dual-tracer approach or if this compound would be useful for theranostic purposes, i.e., to treat tumor burden using a ¹⁷⁷Lu-labeled equivalent (rather not, seeing the low uptake in FDG-avid tumor lesions on [68Ga]Ga-NOTA-T4 immunoPET).

Last, if this paper is of interest for EMBO Molecular Medicine, it may also published as a Short Report.

Referee #1 (Remarks for Author):

The authors investigated [68Ga]Ga-NOTA-T4 immunoPET, an Immuno-PET radiotracer to determine Trophoblast cell surface antigen 2 (Trop2) expression in-vivo. This cell surface antigen serves as a potential pan-cancer biomarker in varying tumor entities. In a translational setting, they reported on heterogeneous expression of Trop2 to identify individuals that may benefit from Trop2-directed therapies. In this regard, the authors used rodents and could demonstrate successful blocking of radiotracer uptake using ABDT4, indicative for specificity. Experiments in a beagle dog provided favorable biodistribution properties with rapid clearance from the blood pool and other organs such as the liver and expected renal excretion. Those experiments were then followed by patients in a head-to-head-comparison with 18F-FDG (reference), most likely to identify tumor heterogeneity.

Comments:

(A) Legend of Figure 4 is misleading. Which animals are presented here? Fig. 4A/B looks like mice, but the authors discuss results on a beagle? ("Figure 4. ABDT4 blocked tumor uptake of [68Ga]Ga-NOTA-T4 and the dynamic distribution patterns of [68Ga]Ga-NOTA-T4 in a beagle dog")

-Preclinical experiments (Fig. 4) clearly show the favorable uptake after ABDT4. Can the authors also provide images without blockage? The provided figures may only depict results after administration of ABDT4. Or are these mice and not a beagle?
-Quantitative analyses (Fig. 4B): Tumor uptake seems to be comparable to uptake in the myocardium after blockage. It would be interesting to have a head-to-head comparison of quantitative results in those large animals without blockage, e.g., by providing a ratio. Fig. 4C: Those ex-vivo results are also after blockage?

Both of those aspects are addressed in Fig. 3 in a dose-dependent manner using mice/murine models, but if such data could be provided for larger animals as provided in Fig. 4, a relevant translational step would have been addressed, as different expression patterns of Trop2 among varying species cannot be ruled out. If such data are not available, this should be discussed in the "Discussion" section. Moreover, in Fig. 4A, the term "blocking Trop2 on the surface of tumor cells with ABDT4" including the arrow should be removed, as in the Material and Methods, the beagle dog seemed not to be infested with tumor cells, but being healthy. This should be clarified, e.g., whether the entire Figure 4 presents data on a beagle or a mouse/mice.

(B) Patients: Patient presented in Fig. 5 appears to be promising, as background radiotracer uptake seems to be low. In Fig. 7, there is reduced uptake in tumor lesions relative to glucose uptake. Please specify whether the benefit of this novel radiotracer is to investigate tumor heterogeneity. The authors also said in their results: "which is a prerequisite for developing molecular imaging or theranostic tracers". If a theranostic usage is planned, can the authors further provide details on labeling their compound with β -emitters? Do the authors see theranostic potential based on tumor uptake in Fig. 5-7?

Referee #2 (Remarks for Author):

The paper tackles a significant clinical problem, proposing an interesting radiopharmaceutical targeting Trop2.

Major changes to the text are suggested:

- The English language should be revised by a native speaker or a dedicated software.
- The introduction is too long and general. This part should be more focused, in particular the paragraphs before the Trop2 description.
- The ABDT4 is first mentioned in the Results, while it should appear in the Introduction also.
- The superiority of T4 compared to T5 should be better explained in the text.
- It is unclear what the dynamic distribution in the eagle dog added to the experiments on mice. It should be commented more in depth.
- The Ethics approval for the first in human use should be better described. The Reviewer understood that the three patients were enrolled in a clinical trial on the basis of Trop2 images. This is difficult to accept, because the Trop2 radiopharmaceutical was experimental.
- The first part of the discussion is useless. It is a repetition of the introduction and a summary of the results. It can be safely reduced and re-focused on the limitations of the study. In particular it is unclear how the authors see the possible applications of this new - and interesting - radiopharmaceutical.

Referee #3 (Remarks for Author):

There is a good reason to determine uptake of the naked antibody and see whether this predicts ADC effects. (Mileva M, et al. Molecular imaging predicts lack of T-DM1 response in advanced HER2-positive breast cancer (final results of ZEPHIR trial). NPJ Breast Cancer. 2024 Jan 6;10(1):4 and Gebhart Ann Oncol 2017. Whether a nanobody is as predictive remains to be proven. Trop2 is an interesting target.

The current limitations of current diagnostic tools are accurately summarized in this section. The need for, preferably non-invasive, diagnostic tools that can effectively deal with interpatient tumor heterogeneity follows, especially in the context of precision medicine and targeted therapy.

To make it clear to the reader what the problem is that Trop2 imaging can solve, it would be nice if the authors could expand on why Trop2 is a promising biomarker. For instance, a short overview of the expression of this antigen and the current issues with

patient selection for Trop2 targeted therapy.

The introduction is too long.

A clear and concise overview highlighting the need for new predictive biomarkers, the latest developments in PET imaging, and how they might help in the discovery of predictive biomarkers. A clear summary of the clinical significance of Trop2 and Trop2-targeted therapies.

What remains elusive from this section of the paper is the potential role of the heterogenous expression of Trop2 in solid malignancies. Is there anything known about the spatial and temporal expression of this antigen? Has IHC been shown to improperly predict therapeutic response to Trop2-targeted therapy?

Results

For the mouse studies, a nice presentation of biodistribution results was given. The paper illustrates the pharmacodynamics/and kinetics of both ...tracers. Tracer biodistribution is well explained.

For the mice study. Tumor-to-organs ratios between blocking doses were described, but not between the scan time point (45 min postinjection and 2.5 hours postinjection). It might be worthwhile to also provide an analysis in order to establish optimal imaging time after tracer administration.

T3M-4 pancreatic cancer cell line is a human CEA + cell line and, therefore, informative regarding tracer behavior gives likely species specificity; if that is the case, biodistribution to normal mouse tissues is not very informative.

Is the study in beagle dogs informative? Is there no species specificity?

For PBS in the methods section is a formula required and pH.

An attempt to establish an optimal tracer dose was established for the mice; however, this was not done when imaging the Beagle or the three patients that were imaged in this study.

The biodistribution of imaging agents can be vastly different between animals and humans. In this study, only three human subjects were imaged. No optimal tracer dose or optimal imaging time was established for the human subjects. Meaning that the results presented in this study need to be interpreted with caution.

For the three human subjects. Tracer uptake in tumor lesions is well described, as well as the relation between tracer tumor uptake and biopsy results for the three patients that entered this study.

Was the tracer GMP produced is there an IMPD?

Has the clinical trial been registered anywhere, like the ClinicalTrials.gov database? This is critically important as it allows the reviewer, etc, to review the design of the trial.

It is not possible to understand Fig 5; statements are made about disappearance, etc, but no information is provided in the pictures.

Did the authors study the whole body distribution of Trop2, e.g., in the human protein atlas, and thereafter check whether these tissues showed uptake?

In the discussion, it is stated that Ga-NOTA-T4 has a cleaner background than FDG-PET; however, in the results, no quantification of background uptake and uptake of major organs in human subjects can be found. However, in the discussion, it is stated that physiologic Trop2 expression can be found in numerous tissue types, and it is suggested that there is a potential link between Trop2 expression in these tissues and potential ADC toxic effects, which the tracer might help predict. According to the discussion, physiologic Trop2 expression was also seen in numerous tissues on the (68Ga)Ga-NOTA-T4 patient scans. A more in-depth analysis and quantification of these findings would be of great interest to help the reader understand the images of the human subjects presented in this article.

Discussion

"Thus, [68Ga]Ga-NOTA-T4 has the potential to noninvasively visualize heterogeneous expression and dynamics of Trop2, improving clinical decision-making when considering Trop2-targeted therapeutics as treatment options."

Based on the data presented in this study, this conclusion is stated too strongly. Further investigation of this tracer based on its safety profile and favorable distribution seems warranted. However, no link between Ga-NOTA-T4 and Trop2-targeted therapy has been established in this study. Nanobody kinetics might behave differently from ADCs, where the toxic payload is attached to the targeting antibody. Therefore, tracer uptake may not be predictive of ADC therapy efficacy.

The references should be checked for adherence to the journal style.

As a service to authors, EMBO provides authors with the possibility to transfer a manuscript that one journal cannot offer to publish to another EMBO publication. The full manuscript and if applicable, reviewers reports are automatically sent to the receiving journal to allow for fast handling and a prompt decision on your manuscript. For more details of this service, and to transfer your manuscript to another EMBO title please click on Link Not Available

Referee #1 (Comments on Novelty/Model System for Author):

The authors investigated [⁶⁸Ga]Ga-NOTA-T4 immunoPET, an Immuno-PET radiotracer to determine Trophoblast cell surface antigen 2 (Trop2) expression in-vivo. This cell surface antigen serves as a potential pan-cancer biomarker in varying tumor entities. In a translational setting, they reported on heterogenous expression of Trop2 to identify individuals that may benefit from Trop2-directed therapies. In this regard, the authors used rodents and could demonstrate successful blocking of radiotracer uptake using ABDT4, indicative for specificity. Experiments in a beagle dog provided favorable biodistribution properties with rapid clearance from the blood pool and other organs such as the liver and expected renal excretion. Those experiments were then followed by patients in a head-to-head-comparison with ¹⁸F-FDG (reference), most likely to identify tumor heterogeneity.

While the herein used approach of investigating mice, followed by large animals and finally humans is promising, the results in large animals are confusing, as the Figure Legends in Figure 4 may be not appropriately labeled. It says "Figure 4. ABDT4 blocked tumor uptake of [68Ga]Ga-NOTA-T4 and the dynamic distribution patterns of [68Ga]Ga-NOTA-T4 in a beagle dog.", but Fig. 4A-B may be mice. This should be clarified. While this radiotracer appears promising in murine models, the uptake in humans appears to be inferior to the reference FDG. As such, the authors should further discuss the value of this radiotracer, e.g., depicting tumor heterogeneity in a dual-tracer approach or if this compound would be useful for theranostic purposes, i.e., to treat tumor burden using a ¹⁷⁷Lu-labeled equivalent (rather not, seeing the low uptake in FDG-avid tumor lesions on [⁶⁸Ga]Ga-NOTA-T4 immunoPET).

Last, if this paper is of interest for EMBO Molecular Medicine, it may also published as a Short Report.

Response: We appreciate the reviewer for sparing time to review our manuscript. We have carefully revised the manuscript following your insightful and valuable comments. Thank you very much.

Referee #1 (Remarks for Author):**Comments:**

(A) Legend of Figure 4 is misleading. Which animals are presented here? Fig. 4A/B looks like mice, but the authors discuss results on a beagle? ("Figure 4. ABDT4 blocked tumor uptake of [⁶⁸Ga]Ga-NOTA-T4 and the dynamic distribution patterns of [⁶⁸Ga]Ga-NOTA-T4 in a beagle dog")

Response: We sincerely apologize for the misleading figure legend. ABDT4 blocking study was carried out only in tumor-bearing mice and total-body dynamic PET/CT imaging was carried out in a healthy beagle dog to investigate the pharmacokinetics of the tracer. Following your comment, we separated ABDT4 blocking group data (Fig. 4 in the revised manuscript) and total-body PET/CT imaging data with the beagle dog (Fig. 5 in the revised manuscript). We have improved the expression, and the revised part of the manuscript has been highlighted in blue to facilitate your reading.

-Preclinical experiments (Fig. 4) clearly show the favorable uptake after ABDT4. Can the authors also provide images without blockage? The provided figures may only depict results after administration of ABDT4. Or are these mice and not a beagle?

Response: Thank you for the nice suggestion. We will add the experimental results without blockage. As mentioned above, the blocking study was carried out in tumor-bearing mice.

-Quantitative analyses (Fig. 4B): Tumor uptake seems to be comparable to uptake in the myocardium after blockage. It would be interesting to have a head-to-head comparison of quantitative results in those large animals without blockage, e.g., by providing a ratio. Fig. 4C: Those ex-vivo results are also after blockage?

Response: We appreciate the reviewer for the constructive comment. We have provided the quantitative analysis results comparing the differential uptake and distribution patterns in the unblocking and blocking groups (Fig. 4D, E in the revised manuscript). Yes, the *ex-vivo* biodistribution study was carried out after the termination of *in-vivo* immunPET/CT imaging studies.

Both of those aspects are addressed in Fig. 3 in a dose-dependent manner using mice/murine models, but if such data could be provided for larger animals as provided in Fig. 4, a relevant translational step would have been addressed, as different

expression patterns of Trop2 among varying species cannot be ruled out. If such data are not available, this should be discussed in the "Discussion" section. Moreover, in Fig. 4A, the term "blocking Trop2 on the surface of tumor cells with ABDT4" including the arrow should be removed, as in the Material and Methods, the beagle dog seemed not be infested with tumor cells, but being healthy. This should be clarified, e.g., whether the entire Figure 4 presents data on a beagle or a mouse/mice.

Response: We apologize that we could not establish tumor-bearing beagle dogs for preclinical studies. We have revised the manuscript following your comments.

(B) Patients: Patient presented in Fig. 5 appears to be promising, as background radiotracer uptake seems to be low. In Fig. 7, there is reduced uptake in tumor lesions relative to glucose uptake. Please specify whether the benefit of this novel radiotracer is to investigate tumor heterogeneity. The authors also said in their results: "which is a prerequisite for developing molecular imaging or theranostic tracers". If a theranostic usage is planned, can the authors further provide details on labeling their compound with β -emitters? Do the authors see theranostic potential based on tumor uptake in Fig. 5-7?

Response: Thank you for the very professional comment. In the registered clinical trial (ClinicalTrials.gov Identifier: NCT06203574), we aim to recruit 60 participants to investigate the safety and diagnostic efficacy of Trop2 immunoPET/CT in patients with solid tumors. Please note that we are still updating the clinical trial registration information because patients with lung cancers are among the candidates for the imaging study.

In theranostic scenario, [^{68}Ga]Ga-NOTA-T4 immunoPET/CT will be used to visualize heterogeneous Trop2 expression and select ones with positive Trop2 expression (such as the patient presented in Fig. 6) for Trop2-targeted therapies, while excluding those with negative Trop2 expression (such as the patient presented in Fig. 8). For therapeutic options, both non-radioactive (such as antibody-drug conjugate) and radioactive (such as radioimmunotherapy) agents can be used. While several Trop2-specific antibody-drug conjugates are readily available for clinical use, radioimmunotherapy strategies targeting Trop2 are still under development. We do

believe that Trop2–targeted radioimmunotherapy may serve as an option for patients with strong Trop2 expression and we are developing Trop2 radioimmunotherapy agents in our lab right now. Following your nice comments, we added relevant discussions in the revised manuscript. Since Trop2 is expressed in several normal organs and tissues, therefore, a blocking dose without payload (either cytotoxic drug or radionuclides) may be necessary to reduce on-target off-tumor side effects when using Trop2-targeted therapeutics. Without Trop2-specific PET imaging approaches, it was very challenging to realize precise patient stratification and response monitoring during Trop2-targeted therapies. Thank you again for all your input and pivotal comments.

Referee #2 (Remarks for Author):

The paper tackles a significant clinical problem, proposing an interesting radiopharmaceutical targeting Trop2.

Response: Thank you very much for the positive feedback on our work developing a Trop2–targeted molecular imaging tracer. With the clinical approval of Sacituzumab Govitecan, Trop2 becomes a target of choice for developing antibody-drug conjugates. We believe that molecular imaging approaches can improve the clinical use of molecularly targeted therapies by precisely selecting suitable patients and dynamically monitoring therapeutic responses [1-3]. Bearing that in mind, we developed the Trop2 tracer [⁶⁸Ga]Ga-NOTA-T4 for clinical use. Currently, we are recruiting more patients with solid tumors to thoroughly explore the diagnostic and predictive value of [⁶⁸Ga]Ga-NOTA-T4 immunoPET/CT.

Major changes to the text are suggested:

- The English language should be revised by a native speaker or a dedicated software.

Response: We have carefully streamlined the expression throughout the manuscript. Thank you for the suggestion.

- The introduction is too long and general. This part should be more focussed, in particular the paragraphs before the Trop2 description.

Response: We have increased the readability of the introduction. Thank you for the

nice suggestion.

- The ABDT4 is first mentioned in the Results, while it should appear in the Introduction also.

Response: Thank you for the professional suggestion. ABDT4 is a T4 derivative that is capable of simultaneously binding to human Trop2 and serum albumin, thus it has a significantly prolonged *in vivo* circulation time. ABDT4 is used to develop nanobody drug conjugates and radioimmunotherapy agents in our lab (unpublished data). In the current study, we exploited ABDT4 to block tumor uptake of [⁶⁸Ga]Ga-NOTA-T4, proving the target specificity of the tracer towards Trop2. We briefly introduced ABDT4 in the revised Introduction part following your comment.

- The superiority of T4 compared to T5 should be better explained in the text.

Response: Thank you for the insightful comment. Previously, we developed a series of Trop2–targeting nanobodies and associated nanobody tracers (*Eur J Nucl Med Mol Imaging*. 2024 Jan;51(2):380-394.). Unfortunately, those nanobodies had a relatively fast dissociation rate after binding to Trop2. Moreover, the radiolabeled nanobody tracers ([⁶⁸Ga]Ga-NOTA-RTD98 and [⁶⁸Ga]Ga-NOTA-RTD01) have high liver accumulation, leading to compromised diagnostic value and limited translational potential. In the submission to *EMBO Molecular Medicine*, we successfully screened second-generation Trop2–targeting nanobodies (T4 and T5) with improved binding curves to recombinant human Trop2 and excellent *in vivo* diagnostic value, thanks to the clear background and high tumor uptake. Moreover, T4 has a higher affinity than T5, which contributed to higher tumor uptake in preclinical imaging studies. Based on the preclinical evidence, we carried out a first-in-human study with [⁶⁸Ga]Ga-NOTA-T4, and initially investigated the safety profiles and differential diagnostic value of the tracer. We introduced the advantages of T4 over T5 in the revised manuscript.

- It is unclear what the dynamic distribution in the eagle dog added to the experiments on mice. It should be commented more in depth.

Response: Thank you for the nice suggestion. We have separated the data in the revised version. The dynamic total-body PET/CT imaging and quantitative analysis

data with the beagle dog are presented in Fig. 5 in the revised manuscript.

- The Ethics approval for the first in human use should better described. The Reviewer understood that the three patients were enrolled in a clinical trial on the basis of Trop2 images. this is difficult to accept, because the Trop2 radiopharmaceutical was experimental.

Response: Thank you for the suggestion. This study was approved by the Institutional Review Board of Huashan Hospital, Fudan University (2023-1017) and registered as a prospective clinical trial (ClinicalTrials.gov Identifier: NCT06203574). Trop2 is a pan-cancer biomarker with varying expression levels in different types of solid tumors. We initially reported data from the three patients in the paper, but we aim to recruit 60 patients with solid tumors to participate in the clinical trial. Please note that we are still updating the clinical trial registration information because patients with lung cancers and other solid tumors are among the candidates for the imaging study.

We hope that the upcoming clinical trial results in relatively large cohorts of patients will provide in-depth information on the diagnostic and predictive value of [⁶⁸Ga]Ga-NOTA-T4 immunoPET/CT. With current evidence, it seems to us [⁶⁸Ga]Ga-NOTA-T4 immunoPET/CT holds promise noninvasively visualizing differential Trop2 expression. The unique value in guiding the use of Trop2-targeted treatments needs to be investigated in the ongoing clinical trial.

- The first part of the discussion is useless. It is a repetition of the introduction and a summary of the results. It can be safely reduced and re-focussed on the limitations of the study. In particular, it is unclear how the authors see the possible applications of this new - and interesting - radiopharmaceutical.

Response: We appreciate the reviewer for the suggestion. We have improved the presentation and the revised contents have been highlighted in blue to facilitate your re-assessment. Thank you again for taking the time to review our work.

Referee #3 (Remarks for Author):

There is a good reason to determine uptake of the naked antibody and see whether this predicts ADC effects. (Mileva M, et al. Molecular imaging predicts lack of T-DM1

response in advanced HER2-positive breast cancer (final results of ZEPHIR trial). NPJ Breast Cancer. 2024 Jan 6;10(1):4 and Gebhart Ann Oncol 2017. Whether a nanobody is as predictive remains to be proven. Trop2 is an interesting target. The current limitations of current diagnostic tools are accurately summarized in this section. The need for, preferably non-invasive, diagnostic tools that can effectively deal with interpatient tumor heterogeneity follows, especially in the context of precision medicine and targeted therapy. To make it clear to the reader what the problem is that Trop2 imaging can solve, it would be nice if the authors could expand on why Trop2 is a promising biomarker. For instance, a short overview of the expression of this antigen and the current issues with patient selection for Trop2 targeted therapy.

Response: We are impressed by the reviewer's understanding of molecular imaging in guiding the development and use of antibody therapeutics. We are long dedicated to the field and are innovating antibody/nanobody-derived diagnostic or theranostic agents for clinical use [1-3]. The clinical success of several antibody-drug conjugates targeting Trop2 indicates the potential of Trop2 as a theranostic biomarker for solid tumors. In this setting, we are exploring full-length antibodies (unpublished data) and nanobodies as targeting vectors to develop radiopharmaceuticals specific for Trop2. Following your nice comments, we have introduced Trop2 expression patterns and the status of patient selection for Trop2-targeted therapies.

The introduction is too long. A clear and concise overview highlighting the need for new predictive biomarkers, the latest developments in PET imaging, and how they might help in the discovery of predictive biomarkers. A clear summary of the clinical significance of Trop2 and Trop2-targeted therapies.

Response: Thank you very much for pointing out the issue. We have streamlined the Introduction part following your comment.

What remains elusive from this section of the paper is the potential role of the heterogenous expression of Trop2 in solid malignancies. Is there anything known about the spatial and temporal expression of this antigen? Has IHC been shown to improperly predict therapeutic response to Trop2-targeted therapy?

Response: Thank you for the insightful comment. Currently, there is still no consensus on the standardized method for determining Trop-2 expression in tumors, which may influence the reproducibility of the results and clinical trials evaluating anti-TROP-2 therapeutics. Consistent with preclinical observations [4], the ASCENT trial showed that triple-negative breast cancer (TNBC) patients with medium-to-high Trop-2 expression achieved double the objective response rate and progression-free survival compared with those with low Trop-2 expression following sacituzumab govitecan treatment [5]. Trop2 is a promising therapeutic target for lung cancers, supported by the results of several basket clinical trials [6]. However, there is no literature reporting the role of Trop2-targeted imaging in refining Trop2-targeted treatments. We hope that our work will provide some clues in this regard.

Results

For the mouse studies, a nice presentation of biodistribution results was given. The paper illustrates the pharmacodynamics/and kinetics of both ...tracers. Tracer biodistribution is well explained. For the mice study. Tumor-to-organs ratios between blocking doses were described, but not between the scan time point (45 min postinjection and 2.5 hours postinjection). It might be worthwhile to also provide an analysis in order to establish optimal imaging time after tracer administration.

Response: Thank you for your valuable suggestion, we have added the quantitative analysis data in the revised manuscript (Appendix Fig. 6). We found that tumor uptake at the two-time points (45 min and 2.5 h) was comparable, but uptake in the circulation (heart), lung, and muscle was significantly lower at the later time point (Appendix Fig. 6A). Therefore, delayed [⁶⁸Ga]Ga-NOTA-T4 immunoPET/CT imaging with clearer background is feasible. At our department, we generally carry out delayed imaging for genitourinary cancers. For patients with genitourinary cancers receiving [⁶⁸Ga]Ga-NOTA-T4 immunoPET/CT imaging, delayed imaging will be added. The clinical benefit of delayed imaging will be reported in our coming work.

T3M-4 pancreatic cancer cell line is a human CEA + cell line and, therefore, informative regarding tracer behavior gives likely species specificity; if that is the

case, biodistribution to normal mouse tissues is not very informative.

Response: Thank you very much for pointing out the issue. We studied the biodistribution of both [^{68}Ga]Ga-NOTA-T4 and [^{68}Ga]Ga-NOTA-T5 probes in normal mice to observe the pharmacokinetics of the probes in normal mice. Probes with better pharmacokinetics are more suitable for subsequent visualization of the tumor-bearing models. In other words, we tend to abandon tracers with bad pharmacokinetics after screening imaging studies with tumor-free mice.

Is the study in beagle dogs informative? Is there no species specificity?

Response: Thank you for the insightful comment. According to regulations in China, it is required to investigate the pharmacokinetics of radiopharmaceuticals before carrying out first-in-human studies. Moreover, the *in vivo* circulation profiles and distribution patterns of radiopharmaceuticals in different species vary a lot. Therefore, we carried out [^{68}Ga]Ga-NOTA-T4 immunoPET/CT imaging using the state-of-the-art EXPLORER total-body PET/CT scanner in an adult male beagle dog (weight approximately 12.5 kg). The good pharmacokinetic data in the beagle dog further warranted the translational potential of [^{68}Ga]Ga-NOTA-T4.

For PBS in the methods section is a formula required and pH.

Response: Thanks for the valuable suggestion, we have added the information in the revised manuscript.

An attempt to establish an optimal tracer dose was established for the mice; however, this was not done when imaging the Beagle or the three patients that were imaged in this study.

Response: Thank you for pointing out this issue. Dose-climbing experiments are required for the implementation of therapeutic drugs to seek the optimal dose, whereas are not required for diagnostic probes in general. In tumor-bearing mice models, we designed the blocking studies with different unlabeled (cold) T4 to see if there is a dose that “reduces kidney accumulation without substantially reducing tumor accumulation of [^{68}Ga]Ga-NOTA-T4”. Unfortunately, we failed to find such an optimal blocking dose to balance tumor uptake and kidney accumulation. As we have discussed in the Discussion part of the manuscript, we are now exploiting other

strategies to reduce kidney accumulation of [⁶⁸Ga]Ga-NOTA-T4. We would like to share with the reviewer that imaging of PEGylated T4 has significantly lower kidney accumulation. We invite the reviewer can keep an eye on our future work in this regard.

The biodistribution of imaging agents can be vastly different between animals and humans. In this study, only three human subjects were imaged. No optimal tracer dose or optimal imaging time was established for the human subjects. Meaning that the results presented in this study need to be interpreted with caution.

Response: Thank you for your insightful advice. Up to now, we have recruited 10 patients to participate in the imaging clinical trial. Following your suggestion, radioactivity uptake of [⁶⁸Ga]Ga-NOTA-T4 in the normal organs and tissues of 10 subjects has been added in the revised manuscript (Appendix Fig. 9). The tracer was injected with an average dose of 130.61 ± 36.04 MBq (n = 10). The safety profile is fair because of the short circulation time of the nanobody tracer in the body, and the low uptake in most of the normal organs. All the subjects did not show any adverse reactions within 4 hours after the probe injection.

For the three human subjects. Tracer uptake in tumor lesions is well described, as well as the relation between tracer tumor uptake and biopsy results for the three patients that entered this study.

Response: Thank you for the nice comment. With accumulating imaging and histopathological data, we will be capable of analyzing the correlation between [⁶⁸Ga]Ga-NOTA-T4 uptake value and Trop2 expression level determined by IHC.

Was the tracer GMP produced is there an IMPD?

Response: Thank you for the professional comment. Yes, all the tracers for clinical use must be produced under GMP conditions. But there are differences between different regulations. As for the approval of radiopharmaceuticals in China, we have an invited paper entitled “Pathway to Approval of Innovative Radiopharmaceuticals in China” nearing publication in the *Journal of Nuclear Medicine*. In the review paper, we introduced in-depth the relevant policies and requirements in China.

Has the clinical trial been registered anywhere, like the ClinicalTrials.gov database?

This is critically important as it allows the reviewer, etc, to review the design of the trial.

Response: We apologize that we did not make this clear in the previous submission. This study was approved by the Institutional Review Board of Huashan Hospital, Fudan University (2023-1017) and registered as a prospective clinical trial (ClinicalTrials.gov Identifier: NCT06203574). We initially reported data from the three patients in the initial submission, but we aim to recruit 60 patients with solid tumors to participate in the clinical trial. In the revised manuscript, we further provided data describing the distribution of [⁶⁸Ga]Ga-NOTA-T4 in normal organs/tissues of 10 candidates (Appendix Fig. 9).

It is not possible to understand Fig 5; statements are made about disappearance, etc, but no information is provided in the pictures.

Response: We apologize that we did not present the medical history of the patient clearly in the initial submission. Following your comment, we provided detailed information about the patient in the supplementary data, so that readers will know the full medical history of the patient. Thank you for the comment.

Did the authors study the whole body distribution of Trop2, e.g., in the human protein atlas, and thereafter check whether these tissues showed uptake?

Response: We apologize that we did not present the information in the previous submission. We provided the results showing the uptake of [⁶⁸Ga]Ga-NOTA-T4 in the normal organs and tissues of 10 subjects (Appendix Fig. 9) . We looked into the whole-body distribution of Trop2 in the public databases human protein atlas (<https://www.proteinatlas.org/ENSG00000184292-TACSTD2/tissue>) and our imaging results correlated well the online sequencing data. In particular, Trop2 is expressed in normal tissues such as the salivary gland, thyroid, and pancreas, we also found obvious uptake in these tissues (Appendix Fig. 9). We also discussed the issue in the revised manuscript.

In the discussion, it is stated that Ga-NOTA-T4 has a cleaner background than FDG-PET; however, in the results, no quantification of background uptake and uptake of major organs in human subjects can be found. However, in the discussion, it is

stated that physiologic Trop2 expression can be found in numerous tissue types, and it is suggested that there is a potential link between Trop2 expression in these tissues and potential ADC toxic effects, which the tracer might help predict. According to the discussion, physiologic Trop2 expression was also seen in numerous tissues on the (68Ga)Ga-NOTA-T4 patient scans. A more in-depth analysis and quantification of these findings would be of great interest to help the reader understand the images of the human subjects presented in this article.

Response: Thank you for the comment. Currently, we have obtained translation data from 10 patients for analysis. We have analyzed the uptake of [⁶⁸Ga]Ga-NOTA-T4 in normal organs and tissues and the result is shown in Appendix Fig. 9 . We have also analyzed the Trop-2 expression patterns in normal organs and tissues. Further analysis with large cohorts of data will be doable when we finish the Trop2-targeted imaging clinical trial.

Discussion

"Thus, [68Ga]Ga-NOTA-T4 has the potential to noninvasively visualize heterogeneous expression and dynamics of Trop2, improving clinical decision-making when considering Trop2-targeted therapeutics as treatment options." Based on the data presented in this study, this conclusion is stated too strongly. Further investigation of this tracer based on its safety profile and favorable distribution seems warranted. However, no link between Ga-NOTA-T4 and Trop2-targeted therapy has been established in this study. Nanobody kinetics might behave differently from ADCs, where the toxic payload is attached to the targeting antibody. Therefore, tracer uptake may not be predictive of ADC therapy efficacy.

Response: Thank you for the valuable comments. As we discussed in the manuscript, we currently have an ADC targeting Trop2 (SHR-A1921 and Sacituzumab Govitecan) undergoing clinical trial in our hospital. The Trop2 imaging clinical trial (ClinicalTrials.gov Identifier: NCT06203574) aims to recruit 60 patients with solid tumors to thoroughly explore and validate the value of [⁶⁸Ga]Ga-NOTA-T4 immunoPET/CT in visualizing Trop2 expression. With enough data in hand, we will first analyze the correlation between tumoral [⁶⁸Ga]Ga-NOTA-T4 uptake and Trop2

expression determined by IHC. For the patients with Trop2–positive tumors entering SHR-A1921 or Sacituzumab Govitecan clinical trials (those are conducted by our oncologist colleagues), we will further analyze the value of [⁶⁸Ga]Ga-NOTA-T4 immunoPET/CT in predicting the treatment responses and evaluating Trop2 dynamics before and after the treatment. Currently, there is no literature reporting clinical-stage Trop2 tracers and their value in patient selection and response evaluation. Following the preliminary success of developing and translating [⁶⁸Ga]Ga-NOTA-T4 for clinical use, now we have a good imaging approach to answer these questions. We invite the reviewer to keep an eye on our upcoming work in this regard. Thank you again for your insightful input.

The references should be checked for adherence to the journal style.

Response: We apologize for the oversight. All the formats and references have been updated following the requirements.

References

1. Wei W, Rosenkrans ZT, Liu J, Huang G, Luo Q-Y, Cai W. ImmunoPET: Concept, Design, and Applications. *Chem Rev.* 2020;120:3787-851. doi:10.1021/acs.chemrev.9b00738.
2. Wu Q, Yang S, Liu J, Jiang D, Wei W. Antibody theranostics in precision medicine. *Med.* 2023;4:69-74. doi:10.1016/j.medj.2023.01.001.
3. Wei W, Younis MH, Lan X, Liu J, Cai W. Single-Domain Antibody Theranostics on the Horizon. *Journal of Nuclear Medicine.* 2022;63:1475-9. doi:10.2967/jnumed.122.263907.
4. Goldenberg DM, Cardillo TM, Govindan SV, Rossi EA, Sharkey RM. Trop-2 is a novel target for solid cancer therapy with sacituzumab govitecan (IMMU-132), an antibody-drug conjugate (ADC). *Oncotarget.* 2015;6:22496-512. doi:10.18632/oncotarget.4318.
5. Bardia A, Tolaney SM, Punie K, Loirat D, Oliveira M, Kalinsky K, et al. Biomarker analyses in the phase III ASCENT study of sacituzumab govitecan versus chemotherapy in patients with metastatic triple-negative breast cancer. *Ann Oncol.* 2021.

doi:10.1016/j.annonc.2021.06.002.

6. Parisi C, Mahjoubi L, Gazzah A, Barlesi F. TROP-2 directed antibody-drug conjugates (ADCs): The revolution of smart drug delivery in advanced non-small cell lung cancer (NSCLC). *Cancer Treat Rev.* 2023;118:102572. doi:10.1016/j.ctrv.2023.102572.

28th Feb 2024

Dear Prof. Wei,

Thank you for submitting your revised manuscript. We have now received the feedback from referees #1 and #2 who who re-reviewed your manuscript. As you will see below, they are satisfied with the revisions. Referee #3 was unfortunately not able to evaluate the revised manuscript, however referees #1 and #2 further assessed your responses to this referee's concerns and stated:

Referee #1:

"I reviewed those comments and I think that comments of referee #3 have been fully addressed."

Referee #2:

"I went through the paper, and I can confirm that all referee's comments were properly addressed."

I will therefore be able to accept your manuscript once the following editorial points will be addressed:

1/ Please address the minor comment from referee #1.

2/Manuscript text:

- Title: we would suggest changing the title to "ImmunoPET imaging of Trop2 in patients with solid tumors".
- We note that you currently have together with you, a total of 3 co-corresponding authors. Is that correct? Do you confirm equal contribution of these 3 people, able to take full responsibility for the paper and its content? While there is no limit per se to the number of co-corresponding authors, 3 is rare, and may not reflect as intended to the community. Please note that an institutional email address is missing for Jiajun Liu, and that ORCID identifiers are currently missing for Jiajun Liu and Fang Xie.
- Please remove the blue text, and only keep in track changes mode any new modification.
- I incorporated minor modifications to your abstract, please let me know if you agree with the following or amend as you see fit:
"Accurately predicting and selecting patients who can benefit from targeted or immunotherapy is crucial for precision therapy. Trophoblast cell surface antigen 2 (Trop2) has been extensively investigated as a pan-cancer biomarker expressed in various tumours, and plays a crucial role in tumorigenesis through multiple signalling pathways. Our laboratory successfully developed two ⁶⁸Ga-labeled nanobody tracers that can rapidly and specifically target Trop2. Of the two tracers, [⁶⁸Ga]Ga-NOTA-T4, demonstrated excellent pharmacokinetics in preclinical mouse models and in a beagle dog. Moreover, [⁶⁸Ga]Ga-NOTA-T4 immuno-positron emission tomography (immunoPET) allowed non-invasive visualization Trop2 heterogeneous and differential expression in preclinical solid tumour models as well as in ten patients with solid tumours. [⁶⁸Ga]Ga-NOTA-T4 immunoPET could facilitate clinical decision-making through patient stratification, as well as response monitoring during Trop2-targeted therapies."
- Please carefully check your text for typos and grammatical mistakes.
- Please remove the figures from the manuscript text.
- "Methods" should be renamed "Materials and Methods"
 - o Production of Trop2 nanobodies: please remove the sentence "The sequence of Trop2-Fc fusion proteins is available upon request" and provide the sequence.
 - o Cells: please indicate whether the cells were authenticated and tested for mycoplasma contamination.
 - o Mice: please provide the housing and husbandry conditions.
 - o Beagle dog: please provide the age of the dog and identify the committee which approved the experiment.
 - o Antibodies: please provide dilutions/concentrations.
 - o Human patients: please provide a sentence stating that the experiments conformed to the principles set out in the WMA Declaration of Helsinki and the Department of Health and Human Services Belmont Report.
 - o Statistics: please provide a statement on sample size, inclusion/exclusion criteria, randomization, and blinding.
- Data Availability section: It is mandatory to include a 'Data Availability' section after the Materials and Methods. Primary datasets produced in this study need to be deposited in an appropriate public database, and the accession numbers and database listed under 'Data Availability'. In case you have no data that requires deposition in a public database, please state so in this section ("This study includes no data deposited in external repositories"). Note that the Data Availability Section is restricted to new primary data that are part of this study.
- Author contributions: CRediT has replaced the traditional author contributions section because it offers a systematic machine-readable author contributions format that allows for more effective research assessment. Please remove the Authors Contributions from the manuscript and use the free text boxes beneath each contributing author's name in our system to add specific details on the author's contribution. More information is available in our guide to authors.
- Please replace "Statements & Declarations" by "Disclosure statement and competing interests". We updated our journal's competing interests policy in January 2022 and request authors to consider both actual and perceived competing interests. Please review the policy <https://www.embopress.org/competing-interests> and update your competing interests if necessary.
- Figure legends should be placed after the references.

3/ Figures and Appendix:

- Please provide exact p values, not a range, in the figure or their legends.
- Please remove the figures from the manuscript and upload them as individual files. The legends should be in the manuscript, after the references.
- Please note that we replaced Supplementary Information with Expanded View (EV) Figures and Tables that are collapsible/expandable online. A maximum of 5 EV Figures can be typeset. EV Figures should be cited as 'Figure EV1, Figure EV2' etc... in the text and their respective legends should be included in the main text after the legends of regular figures. For the figures that you do NOT wish to display as Expanded View figures, they should be bundled together with their legends in a single PDF file called *Appendix*, which should start with a short Table of Content. Appendix figures should be referred to in the main text as: "Appendix Figure S1, Appendix Figure S2" etc.
- Please rename the video "Movie EV", and the callout corrected in the manuscript text. A legend should be zipped with the movie file.
- Please make sure that all figures/figure panels are referenced in the text (callouts are currently missing for Fig. 6 B, D, E, F).
- Please address the following queries from our data editors in the figure legends:
 1. Please note that a separate 'Data Information' section is required in the legends of figures 2b-c, e-f; 3e-g; 4d-e; 6a-f.
 2. Please note that the legends for figures 1b-e are not provided in the sequential manner (legend for figure 1d is provided before legend of figure 1b-c, legend for figure 1e is provided before legend of figure 1c). This needs to be rectified.
 3. Please note that the legends for figures 2b-e are not provided in the sequential manner (legend for figure 2d is provided before legend of figure 2b-c, legend for figure 2e is provided before legend of figure 2c). This needs to be rectified.
- Figure re-use should be mentioned in the figure legends (i.e. Figure 1D and Appendix Figure 2A, Figure 6 and Appendix Figure 10D, Figure 7A/B and Appendix Figure S12).

4/ At EMBO Press we ask authors to provide source data for the main figures. Our source data coordinator will contact you to discuss which figure panels we would need source data for and will also provide you with helpful tips on how to upload and organize the files.

5/ Please provide a complete author checklist, which you can download from our author guidelines (<https://www.embopress.org/page/journal/17574684/authorguide#submissionofrevisions>). Please insert information in the checklist that is also reflected in the manuscript. The completed author checklist will also be part of the RPF.

6/ I introduced minor modifications to your Paper Explained, please let me know if you agree with the following or amend as you see fit:

Problem

Diagnostic imaging methods for malignant tumors such as computed tomography (CT) and magnetic resonance imaging (MRI) are currently unable to capture changes at the molecular or cellular levels. Invasive histological biopsy faces inherent limitations and detection biases due to the heterogeneous nature of the tumors. There is thus an urgent need for noninvasive techniques to visualize the heterogeneous expression of biomarkers such as trophoblast cell surface antigen 2 (Trop2), a promising oncogenic pan-cancer biomarker.

Results

Two novel nanobody tracers (i.e., [68Ga]Ga-NOTA-T4 and [68Ga]Ga-NOTA-T5) with high binding affinity were developed to specifically target Trop2. Preclinical studies in tumor-bearing murine models revealed the high pharmacokinetics and diagnostic potentials of these tracers. [68Ga]Ga-NOTA-T4 further showed excellent pharmacokinetics in a beagle dog, warranting translational potential. In three patients with solid tumors, [68Ga]Ga-NOTA-T4 immuno-positron emission tomography (immunoPET) noninvasively and precisely visualized negative, positive, and heterogeneous Trop2 expression in primary and metastatic tumors, which was confirmed by immunohistochemistry staining.

Impact

The two novel Trop2-targeted nanobody tracers exhibit promising diagnostic value, and could allow identification of patients who would benefit from Trop2-targeted therapies and evaluation of post-treatment therapeutic responses.

7/ Every published paper now includes a 'Synopsis' to further enhance discoverability. Synopses are displayed on the journal webpage and are freely accessible to all readers. They include a short stand first (maximum of 300 characters, including space) as well as 2-5 one-sentences bullet points that summarizes the paper. Please write the bullet points to summarize the key NEW findings. They should be designed to be complementary to the abstract - i.e. not repeat the same text. We encourage inclusion of key acronyms and quantitative information (maximum of 30 words / bullet point). Please use the passive voice. Please attach these in a separate file or send them by email, we will incorporate them accordingly.

8/ As part of the EMBO Publications transparent editorial process initiative (see our Editorial at <http://embomolmed.embopress.org/content/2/9/329>), EMBO Molecular Medicine will publish online a Review Process File (RPF) to accompany accepted manuscripts.

This file will be published in conjunction with your paper and will include the anonymous referee reports, your point-by-point response and all pertinent correspondence relating to the manuscript. Let us know whether you agree with the publication of the RPF and as here.

I look forward to receiving your revised manuscript.

Yours sincerely,

Lise Roth

Lise Roth, PhD

Senior Editor

EMBO Molecular Medicine

***** Reviewer's comments *****

Referee #1 (Remarks for Author):

Blocking studies have now been provided in details. There is a missing reference: "We assume that clinical trial enrollment criteria for Trop2 ADC therapy were not based exclusively on Trop2-positive expression and thus did not result in highly satisfactory treatment efficacy."

Referee #2 (Remarks for Author):

Suitable for publication.

The authors addressed the minor editorial issues.

7th Mar 2024

Dear Prof. Wei,

Thank you for submitting your revised files. Almost everything is fine now, however we would like you to address the following before final acceptance:

- In the methods, statistics, please provide a statement on sample size, inclusion/exclusion criteria, randomization, and blinding.
- Thank you for providing exact p values. Please also provide exact p values for Figure EV1.
- Please address the following query from our data editors: a separate 'Data Information' section is required in the legends of figures 2b-c, e-f; 3e-g; 4d-e; 6a-f.
- Checklist:
 - o Please check the section Cell materials/primary cultures, as I don't think it applies to your study.
 - o Please check the section Experimental animals/Animal observed in or captured from the field, as I don't think it applies to your study.
 - o Please complete the Experimental study design and statistics section (randomization, blinding, inclusion/exclusion criteria)
- Figure re-use: thank you for clarifying. Could you please provide source data for Appendix Figure 2A and Figure EV4 (previously Appendix Figure S12)?
- Thank you for providing a synopsis text. Please change the format to the following:
 - o Stand-first (max. 300 characters, including spaces)
 - o 2 to 5 bullet points (max. 30 words each)
- Thank you for providing a nice synopsis picture. Could you please upload a similar picture with a white background?

Looking forward to receiving your revised files.

Yours sincerely,

Lise Roth

The authors addressed the remaining editorial issues.

12th Mar 2024

Dear Prof. Wei,

Thank you for providing the revised files. I am pleased to inform you that your manuscript is accepted for publication and is now being sent to our publisher to be included in the next available issue of EMBO Molecular Medicine.

With kind regards,

Lise Roth
